# Geometric Image Editing via Effects-Sensitive In-Context Inpainting with Diffusion Transformers

Shuo Zhang[1][*]  Wenzhuo Wu[1][*]  Huayu Zhang[2]  Jiarong Cheng[2,3]  Xianghao Zang[2]  Chao Ban[2]
Hao Sun[2]  Zhongjiang He[2][†]  Tianwei Cao[1][‡]  Kongming Liang[1][‡]  Zhanyu Ma[1]
[1]School of Artificial Intelligence, Beijing University of Posts and Telecommunications
[2]Institute of Artificial Intelligence (TeleAI), China Telecom  [3]Beijing Institute of Technology
{zhangshuo123, wuwenzhuo, caotianwei, liangkongming, mazhanyu}@bupt.edu.cn
{zhanghy56, zangxh, banc}@chinatelecom.cn
{lf_cyan27}@outlook.com  {sun.010, hezhongj_1}@163.com

## Abstract

Recent advances in diffusion models have significantly improved image editing. However, challenges persist in handling geometric transformations, such as translation, rotation, and scaling, particularly in complex scenes. Existing approaches suffer from two main limitations: (1) difficulty in achieving accurate geometric editing of object translation, rotation, and scaling; (2) inadequate modeling of intricate lighting and shadow effects, leading to unrealistic results. To address these issues, we propose GeoEdit, a framework that leverages in-context generation through a diffusion transformer module, which integrates geometric transformations for precise object edits. Moreover, we introduce Effects-Sensitive Attention, which enhances the modeling of intricate lighting and shadow effects for improved realism. To further support training, we construct RS-Objects, a large-scale geometric editing dataset containing over 120,000 high-quality image pairs, enabling the model to learn precise geometric editing while generating realistic lighting and shadows. Extensive experiments on public benchmarks demonstrate that GeoEdit consistently outperforms state-of-the-art methods in terms of visual quality, geometric accuracy, and realism.

## 1 Introduction

Image editing has achieved remarkable progress with recent advancements in generative models, enabling a wide range of practical applications (Huang et al., 2025b). Nevertheless, it remains a challenging task to handle geometric transformations, where an object within an image is translated, rotated, or scaled while preserving scene coherence. The challenge becomes more significant when dealing with large transformations (e.g. long-distance translations, large-angle rotations, and significant scaling) or complex scenes.

This task, commonly termed *geometric image editing*, aims to perform geometric transformations on objects within an image while maintaining object-background consistency during transformation. In this editing task, two key challenges remain: (1) accurate geometric transformations, including translation, rotation, and scaling; (2) natural blending with proper visual effects (e.g., shadows or reflections). Early methods typically relied on copying and pasting objects into target locations, followed by image harmonization (Wang et al., 2024). While simple, such strategies struggle with large transformations and fail to produce realistic lighting and shadow effects. Recent advances in diffusion models have enabled geometry-aware editing by inverting images into noise space and applying affine transformations before decoding (Pandey et al., 2024; Sajnani et al., 2025; Zhu et al., 2025). Although these approaches support a broader range of transformations, they still lack

---

[*]Equal contribution.

[†]Project lead.

[‡]Corresponding author.

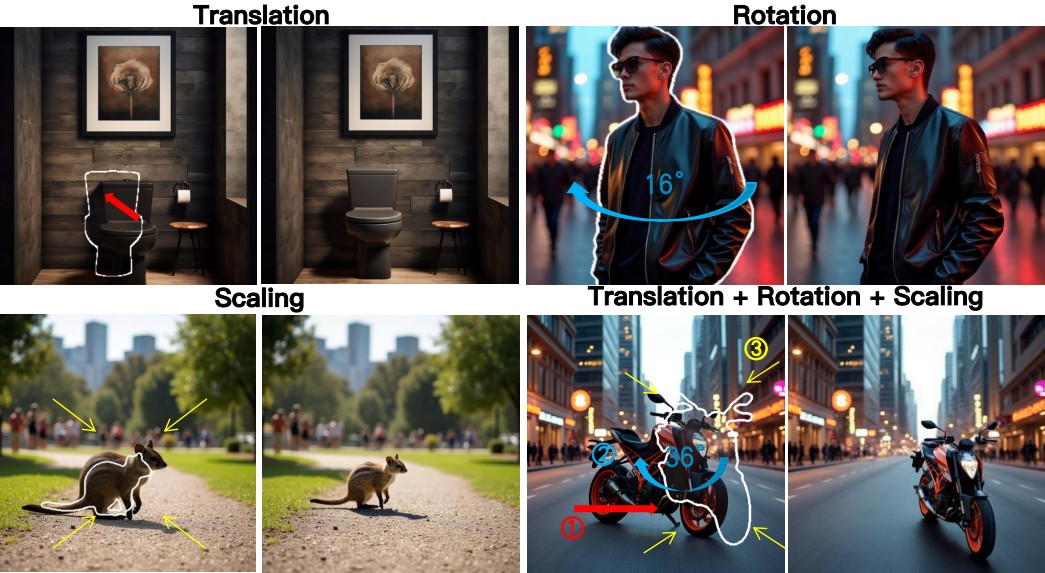

Figure 1: Our method accurately performs geometric edits including translation, rotation, scaling, and their combinations (e.g., translation combined with rotation and scaling), while achieving reliable generation of lighting and shadow effects to ensure realistic editing results.

physically consistent illumination and shadows. A complementary line of work leverages large-scale video datasets to learn environmental lighting priors (Alzayer et al., 2025; Yu et al., 2025; Cheng et al., 2025a). However, these methods remain limited in achieving precise and complex object geometry transformations. Taken together, existing approaches have yet to simultaneously achieve high-fidelity object transformations and photorealistic visual effects.

To address these challenges, we present **GeoEdit**, an effects-sensitive in-context inpainting framework built upon diffusion transformers (Peebles & Xie, 2023). More specifically, to address precise object transformations, we introduce **Geometric Transformation** module that employs 3D reconstruction to lift objects into an elevated dimensional space, where parametric transformations (translation/rotation/scaling) are executed with precise control. To address the realistic generation of lighting and shadow effects, we design **Effects-Sensitive Attention** (ESA) that refines the prior distribution of the attention map, guiding the post-training model to generate more plausible visual effects. Meanwhile, the superiority of ESA is also supported by theoretical analysis (see Thm 3.1).

Unfortunately, existing datasets fail to simultaneously provide precise geometric transformations and high-quality lighting/shadow effects (Peebles & Xie, 2023; Yu et al., 2025; Alzayer et al., 2025; Cheng et al., 2025b), leading them to be inadequate for training our model. To fill this gap, we construct **RS-Objects**, which is a large-scale dataset containing 120,000 image pairs (20,000 rendered and 100,000 synthetic) spanning 30 objects across 24 scenes. This ensures adequate training of our proposed model.

In light of these designs, **GeoEdit** enables precise geometric transformation and faithful visual effects (see Figure 1). To summarize, our main contributions are threefold:

1. We propose **GeoEdit**, an in-context inpainting framework that integrates a **Geometric Transformation** module for precise object editing and **Effects-Sensitive Attention** for realistic generation, with the effectiveness of ESA also theoretically supported.

2. We construct **RS-Objects**, a large-scale geometric editing training dataset containing over 120,000 samples, covering 30 different object categories and 24 complex scenes.

3. Through extensive experiments, we demonstrate that **GeoEdit** achieves superior results and outperforms existing methods.

## 2 RELATED WORK

**Diffusion Models.** Diffusion models synthesize images by iteratively denoising noisy samples with a learned noise estimator (Ho et al., 2020; Song et al., 2020b). Significant improvements have been achieved in many aspects such as efficient sampling (Song et al., 2020a), latent space modeling (Rombach et al., 2022; Nichol et al., 2021), and guidance mechanisms (Ho & Salimans, 2022; Dhariwal & Nichol, 2021). Notably, recent development of Diffusion Transformers (Peebles & Xie, 2023) has demonstrated superior generation capabilities and has been adopted by many advanced image generation models. In this paper, we explore utilizing diffusion transformers to tackle geometric editing.

**Geometric Image Editing.** Recent geometric image editing methods can be categorized by their need for per-instance training. *Training-free* approaches impose geometric constraints during inference on pre-trained generative models. These approaches typically either (i) optimize latent features to match user control points (Pan et al., 2023), or (ii) manipulate diffusion features and attention maps to reflect geometric transformations (Pandey et al., 2024; Sajnani et al., 2025; Zhu et al., 2025). Realism is often enhanced via post-processing modules for harmonization or relighting (Tsai et al., 2017; Cong et al., 2020), with recent work integrating diffusion-based harmonization and counterfactual supervision for physical plausibility (Song et al., 2023; Winter et al., 2024; Kim et al., 2025). However, these methods often require auxiliary inputs and can produce artifacts under significant pose changes. *Training-based* editors learn task-specific geometric priors via (i) *test-time fine-tuning* with lightweight adaptation (Shi et al., 2024; Zhang et al., 2025b; Shi et al., 2025), or (ii) *supervised learning* on video/3D datasets to distill physical priors (Alzayer et al., 2025; Yu et al., 2025; Cheng et al., 2025b; Yenphraphai et al., 2024; Wu et al., 2024; Michel et al., 2023). In contrast, **GeoEdit** does not rely on per-instance fine-tuning; it performs in-context DiT inpainting guided by texture information as well as geometric guidance.

**In-Context Learning for Image Editing.** Visual in-context learning treats references, exemplar pairs, and controls as a unified visual prompt, processed in a single forward pass. In image editing, this paradigm powers paint-by-example approaches that preserve appearance via cross-attention or lightweight adapters (Yang et al., 2023; Ye et al., 2023), and panel-based layouts that facilitate few-shot adaptation (Wang et al., 2023b; Chen et al., 2023; Zhang et al., 2025a; Chen et al., 2025). These methods have been extended to mask-conditioned object insertion on DiT architectures (Song et al., 2025) and broader general-purpose frameworks (Bar et al., 2022; Wang et al., 2023a). **GeoEdit** also adopts this framework, while introducing several task-specific modifications to better address the challenges of geometric editing.

## 3 GEOEDIT

Our goal is to achieve precise geometric editing while generating realistic lighting and shadow effects. To this end, we propose **GeoEdit**, a diffusion-based framework for geometric image editing (see Figure 2). Given an input image and a source mask, our **Geometric Transformation** module (Section 3.1) applies translation, rotation, and scaling to produce a target mask and an appearance reference (transformed object) for in-context guidance. These inputs, together with the original image, are processed by the Diffusion Transformer module, where paired masks explicitly constrain content generation and the **Effects-Sensitive Attention** (ESA) (Section 3.2) mechanism adaptively captures lighting and shadow effects to produce realistic editing results. The resulting representations are then decoded to produce the final edited image.

### 3.1 GEOMETRIC TRANSFORMATION

To better achieve precise geometric editing of objects, we model Geometric Transformation (shown in the bottom-left panel of Figure 2) as translation, rotation and scaling. We precompute object images and masks under these transformations to provide geometric and texture cues for in-context reasoning and downstream editing. Each operation preserves object appearance while enabling precise geometric control.

**Translation.** We copy the source mask to the target location to reposition the object without altering its shape or texture, providing a stable spatial reference for subsequent transformations.

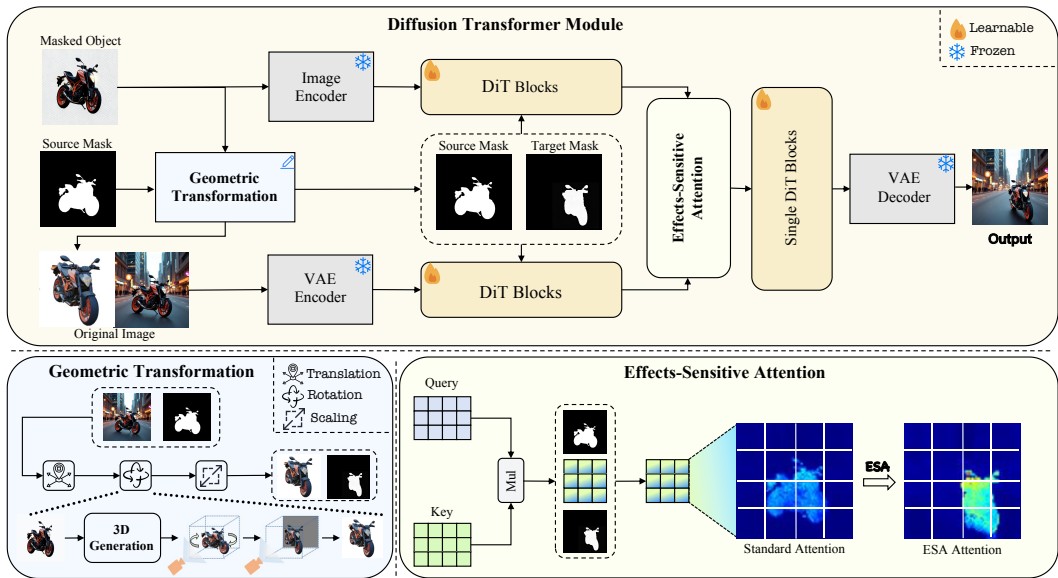

Figure 2: The framework of proposed GeoEdit, built upon an in-context inpainting paradigm, consists of a Diffusion Transformer Module that integrates two key components: (1) Geometric Transformation for object editing (translation, rotation, and scaling), and (2) Effects-Sensitive Attention for modeling intricate lighting and shadow effects.

**Rotation.** We reconstruct each object as a textured 3D mesh using Hunyuan3D-2.1 (Hunyuan3D et al., 2025) and rotate it to arbitrary angles. The mesh is orthographically projected onto a white canvas. To avoid clipping, we first render on a canvas three times larger than the target resolution and use a depth buffer for occlusion. The projected region is cropped to the object's bounding box and rescaled with a safety factor of $0.7$ to fit the target resolution while preserving aspect ratio. The content is then centered. A corresponding mask is generated by rendering the mesh in white on a black background using the same procedure. Finally, the transformed object and corresponding mask are translated to the target location.

**Scaling.** We simulate depth variation by uniformly scaling the object image and mask according to the intended camera-axis displacement, providing simple depth cues and completing the geometry-aware input preparation with translation and rotation.

## 3.2 EFFECTS-SENSITIVE ATTENTION

In geometric editing, the modeling of intricate lighting and shadow effects is often insufficient. To address this, we explore different modulation strategies for guiding attention and propose **Effects-Sensitive Attention** (ESA), a soft guidance mechanism that biases attention according to regional objectives while preserving cross-region interactions.

We first consider the **standard attention** for object insertion in geometric editing, where the query of token $i$ is denoted as $q_i \in \mathcal{T}^{(Q)}$ and the key of token $j$ as $k_j \in \mathcal{T}^{(K)}$. Here, $\mathcal{T}^{(Q)}$ represents the set of tokens corresponding to all regions of the image, while $\mathcal{T}^{(K)}$ represents the set of tokens encoding object-specific features. The interaction between $q_i \in \mathcal{T}^{(Q)}$ and $k_j \in \mathcal{T}^{(K)}$ can then be expressed as the scaled dot-product similarity:

$$A_{ij} = g(S_{ij}), \quad S_{ij} = \frac{q_i k_j^\top}{\sqrt{d}}, \tag{1}$$

where $S_{ij}$ denotes the similarity between query $i$ and key $j$, $d$ denotes the dimensionality of the query and key vectors, $g(S_{ij}) = \exp(S_{ij})/\sum_i \exp(S_{ij})$ denotes the softmax function. This formulation distributes attention broadly across the entire scene, which helps maintain global context.

However, this broad distribution often means that the editing region lacks sufficient focus. As shown in Figure 3, standard attention struggles to accurately integrate objects within the editing regions.

One straightforward way to modulate attention is to only focus certain queries on a subset of keys (Sun et al., 2025). We define $\mathcal{T}^{(Q)}\mathrm{edit}$ as the set of tokens corresponding to insertion regions, while $\mathcal{T}^{(Q)}\mathrm{aux}$ denotes the tokens in the other regions, such that $\mathcal{T}^{(Q)} = \mathcal{T}^{(Q)}\mathrm{edit} \cup \mathcal{T}^{(Q)}\mathrm{aux}$. Accordingly, the **Hard Modulation** can be written as:

$$A_{ij}^{\mathrm{hard}} = g\big(S_{ij}^{\mathrm{hard}}\big), \quad S_{ij}^{\mathrm{hard}} = \begin{cases} +\infty, & q_i \in \mathcal{T}_{edit}^{(Q)}, \\ q_i k_j^\top / \sqrt{d}, & q_i \in \mathcal{T}_{aux}^{(Q)}, \end{cases} \tag{2}$$

However, since the auxiliary regions include lighting and shadow effects, suppressing attention from these regions to the object may result in missing these effects. As shown in Figure 3, Hard Modulation can insert objects, but it fails to properly render the associated shadows and lighting effects.

To enhance attention within the insertion region by focusing on object tokens, while preserving interactions with auxiliary regions to capture lighting and shadow effects. We propose **Effects-Sensitive Attention** (ESA). The equation of ESA can be written as follows:

$$A_{ij}^{\mathrm{ESA}} = g\big(S_{ij}^{\mathrm{ESA}}\big), \quad S_{ij}^{\mathrm{ESA}} = \begin{cases} q_i k_j^\top / \sqrt{d} + \delta, & q_i \in \mathcal{T}_{edit}^{(Q)}, \\ q_i k_j^\top / \sqrt{d}, & q_i \in \mathcal{T}_{aux}^{(Q)}, \end{cases} \tag{3}$$

Here, $\delta = \alpha \cdot \mathrm{std}(S_{ij})$ denotes a scaled standard deviation of the raw attention logits, where the scaling factor $\alpha > 0$ represents the strength of control over attention. As shown in Figure 3, ESA can insert objects while maintaining high-quality shadows and lighting.

While the previous discussion focused on object insertion, geometric editing involves both object insertion and background restoration. Similar to Equation 3, we enhance the attention of tokens in the background restoration regions to the background features. The hyperparameter $\alpha$ is set differently for object insertion and background restoration, with specific experimental details provided in the Appendix (see Appendix G)).

In addition to aforementioned analysis, Theorem 3.1 further reveal that ESA can align the prior distribution of the attention map with an ideal attention structure $A^\star$. Since that $A^\star$ simultaneously attends to object and visual effect regions (see Appendix A.1), the alignment with $A^\star$ may boost visual effect generation for post-training models. The full proof process refers to Appendix A.

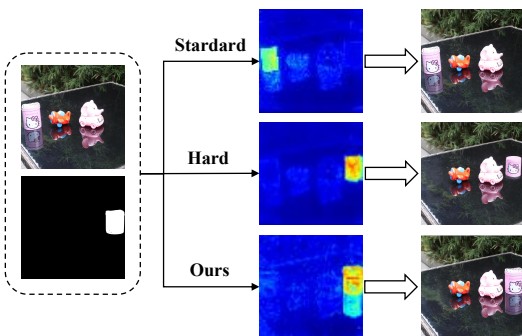

Figure 3: Comparison of attention strategies: standard; Hard Modulation; Ours.

**Theorem 3.1** *Let $A^\star$ be an ideal attention map, and $\rho$ be its threshold for discriminating critical/non-critical regions. Here $A^\star$ has several necessary conditions defined in Appendix A.1. Based on this, if we have $\rho \geq 1/|\mathcal{T}_{\mathrm{edit}}^{(Q)}|$, then the following statements hold for each key token $k_j \in \mathcal{T}^{(K)}$:*

1. *The subtraction $D_{\mathrm{KL}}\big(A_{\cdot j}^\star \,\|\, A_{\cdot j}\big) - D_{\mathrm{KL}}\big(A_{\cdot j}^\star \,\|\, A_{\cdot j}^{\mathrm{ESA}}\big) \geq \delta(|\mathcal{T}_{\mathrm{edit}}^{(Q)}| \cdot \rho - 1) \geq 0$, thus we have $D_{\mathrm{KL}}\big(A_{\cdot j}^\star \,\|\, A_{\cdot j}^{\mathrm{ESA}}\big) \leq D_{\mathrm{KL}}\big(A_{\cdot j}^\star \,\|\, A_{\cdot j}\big)$.*

2. *The divergence $D_{\mathrm{KL}}\big(A_{\cdot j}^\star \,\|\, A_{\cdot j}^{\mathrm{hard}}\big) \to +\infty$ and $D_{\mathrm{KL}}\big(A_{\cdot j}^\star \,\|\, A_{\cdot j}^{\mathrm{ESA}}\big)$ has a finite upper bound $D_{\mathrm{KL}}\big(A_{\cdot j}^\star \,\|\, A_{\cdot j}^{\mathrm{ESA}}\big) \leq |\mathcal{T}^{(Q)}| \log\big(1/\min(A_{ij})\big)$, thus we have $D_{\mathrm{KL}}\big(A_{\cdot j}^\star \,\|\, A_{\cdot j}^{\mathrm{ESA}}\big) \leq D_{\mathrm{KL}}\big(A_{\cdot j}^\star \,\|\, A_{\cdot j}^{\mathrm{hard}}\big)$.*

where $D_{\text{KL}}$ denotes KL divergence; $A^\star_{\cdot j}$ denotes the attention distribution w.r.t $A^\star$ over each query token $q_i \in \mathcal{T}^{(Q)}$; $A^{\text{hard}}_{\cdot j}$, $A^{\text{ESA}}_{\cdot j}$ and $A_{\cdot j}$ are defined by following the same paradigm with $A^\star_{\cdot j}$.

In this way, ESA maintains attention flow across the entire scene while softly biasing attention toward relevant tokens. As shown in Figure 2, this approach increases the attention of editing regions, while still preserving interactions with auxiliary regions.

# 4 DATASET CONSTRUCTION PIPELINE

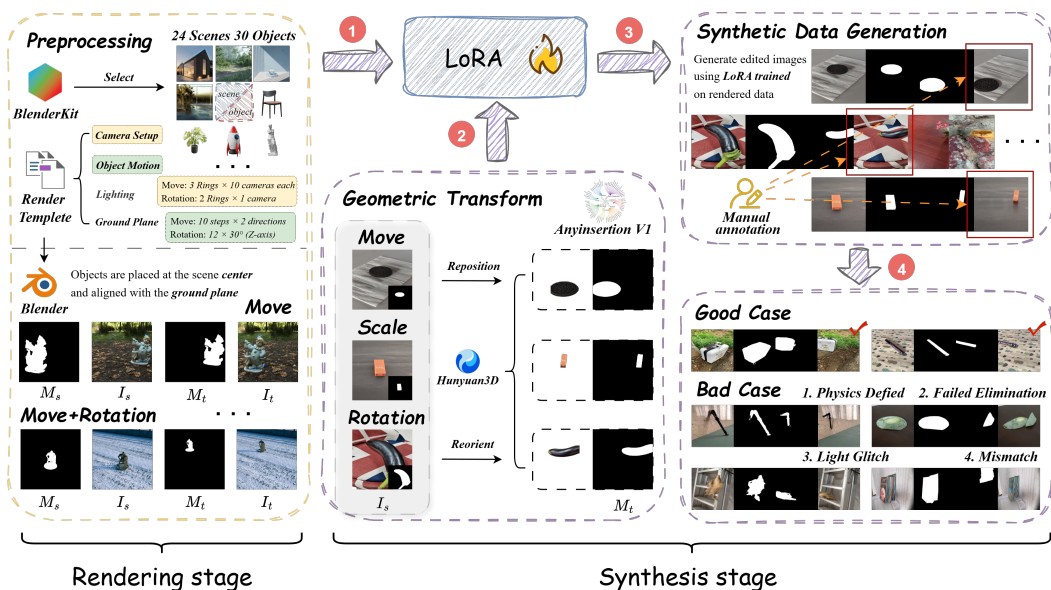

Figure 4: The rendering-synthesis pipeline for building our RS-Objects dataset.

Collecting large-scale images of objects with precisely controlled transformations while preserving realistic lighting and shadow effects is inherently challenging. To overcome this, we introduce **RS-Objects**, a dataset constructed through a two-stage *rendering-synthesis* strategy. This approach provides fully controllable training samples exhibiting realistic visual effects, while helping bridge the domain gap between synthetic and real images. The overall pipeline is shown in Figure 4.

**Rendering Stage.** In step (1) of the process (shown in Figure 4), we use Blender (Community, 2018) to render 24 diverse object-rich scenes with 30 distinct objects. Multiple camera rings are employed under parameterized translation, rotation, and scaling, yielding an extensive Rendered Dataset of 20,000 image pairs. This dataset captures detailed object geometry and appearance (see visualization in Appendix B), which is used to train the initial LoRA (Hu et al., 2022).

**Synthesis Stage.** This stage comprises three main steps (shown in the right part of Figure 4): (2) Mesh-based sample generation, (3) Large-scale image synthesis using the trained LoRA, and (4) Human-in-the-loop quality filtering. Specifically, step (2) leverages meshes from AnyInsertion_-V1 (Song et al., 2025) and Hunyuan3D 2.1 (Hunyuan3D et al., 2025) to generate preprocessed images and target masks, enriching the dataset with diverse geometry and textures. Step (3) employs the LoRA model trained on the Rendered dataset for batch generation of geometry- and texture-aware object images, producing over 800,000 synthesized samples for robust downstream training. Step (4) involved a 20-person annotation team conducting a three-week quality assessment to discard samples with issues such as spatial coherence, feature consistency, illumination consistency, and controlled generation. After this rigorous filtering (see Appendix C), we retained over 100,000 high-quality image–mask pairs constituting the final AIGC Dataset for training.

The final **RS-Objects** dataset is constituted by the Rendered Dataset and the AIGC Dataset, containing a total of **over 120,000 high-quality rendered and synthesized image-mask pairs** that exhibit precise geometric transformations and realistic lighting, shadow effects.

Table 1: Quantitative results on 2D-edits and 3D-edits tasks. We report seven metrics for image quality, consistency, and editing effectiveness. Best results are in bold, second best are underlined.

| Methods | Editing Tasks | FID↓ | DINOv2↓ | KD↓ | SUBC↑ | BC↑ | WE↓ | MD↓ |
|---|---|---|---|---|---|---|---|---|
| RegionDrag Lu et al. (2024) | | 41.88 | 257.43 | 0.052 | 0.796 | 0.972 | 0.120 | 32.75 |
| MotionGuidance Geng & Owens (2024) | | 146.41 | 1307.90 | 0.078 | 0.452 | 0.714 | 0.260 | 145.46 |
| DragDiffusion Shi et al. (2024) | | 37.68 | 242.52 | 0.051 | 0.776 | 0.969 | 0.177 | 34.78 |
| Diffusion Handles Pandey et al. (2024) | | 69.34 | 588.58 | 0.054 | 0.725 | 0.857 | 0.180 | 40.94 |
| GeoDiffuser Sajnani et al. (2025) | 2D-edits | 38.22 | 198.58 | 0.052 | 0.761 | 0.937 | 0.166 | 34.94 |
| DesignEdit Jia et al. (2024) | | 32.55 | 142.45 | 0.052 | 0.874 | 0.962 | 0.098 | 10.15 |
| Magic Fixup Alzayer et al. (2025) | | 27.32 | 114.08 | 0.051 | 0.889 | 0.966 | 0.075 | 10.39 |
| FreeFine Zhu et al. (2025) | | 27.48 | 109.23 | 0.052 | 0.906 | 0.971 | 0.056 | 9.42 |
| **GeoEdit (Ours)** | | **25.07** | **90.66** | **0.051** | **0.910** | **0.977** | **0.054** | **9.23** |
| Diffusion Handles Pandey et al. (2024) | | 126.24 | 1028.60 | 0.056 | 0.737 | 0.885 | 0.189 | 18.56 |
| GeoDiffuser Sajnani et al. (2025) | 3D-edits | 77.34 | 475.62 | 0.055 | 0.802 | 0.946 | 0.179 | 43.51 |
| FreeFine Zhu et al. (2025) | | 65.94 | 366.39 | 0.055 | 0.832 | 0.967 | 0.052 | 21.27 |
| **GeoEdit (Ours)** | | **64.30** | **350.69** | **0.054** | **0.840** | **0.977** | **0.051** | **18.08** |

## 5 EXPERIMENTS

### 5.1 EXPERIMENTAL DETAILS

**Implementation Details. GeoEdit** builds upon FLUX.1 Fill (Black Forest Labs, 2024), an inpainting model based on the DiT architecture. We replace the original T5 text encoder (Raffel et al., 2020) with a SigLIP image encoder (Zhai et al., 2023) and fine-tune it using LoRA (Hu et al., 2022). All training and inference procedures follow standard practices with high-resolution images. Detailed training configurations and hyperparameters are provided in Appendix D.

**Comparison Methods.** We benchmark our framework against representative image-editing methods under different geometric transformations. For 2D-edits, we compare with RegionDrag (Lu et al., 2024), MotionGuidance (Geng & Owens, 2024), DragDiffusion (Shi et al., 2024), Diffusion-Handles (Pandey et al., 2024), GeoDiffuser (Sajnani et al., 2025), DesignEdit (Jia et al., 2024), Mag-icFixUp (Alzayer et al., 2025), and FreeFine (Zhu et al., 2025). For 3D-edits, we evaluate against methods explicitly modeling 3D transformations—DiffusionHandles, GeoDiffuser, and FreeFine. This division reflects the different challenges of 2D versus 3D editing and enables targeted evaluation.

**Datasets and Evaluation Metrics.** We evaluate on GeoBench (Zhu et al., 2025), which integrates PIE-Bench (Ju et al., 2024) and Subjects200K (Tan et al., 2024), yielding 811 source images and 5,988 editing instructions spanning 2D (translation, scaling) and 3D (rotation) tasks. Following FreeFine (Zhu et al., 2025), we adopt seven metrics: (1) **Image Quality:** Fréchet Inception Distance (FID) (Heusel et al., 2017) computed separately per task type, plus Kernel Distance (KD) (Binkowski et al., 2018) and DINOv2 feature distance (Stein et al., 2023). (2) **Consistency:** Subject Consistency (SUBC) and Background Consistency (BC) measure similarity of foreground and background embeddings, using source mask and target mask. (3) **Editing Effectiveness:** Warp Error (WE) and Mean Distance (MD) quantify how closely generated objects match target configurations within masked regions. Together these metrics provide complementary views of realism, consistency, and adherence to editing instructions.

### 5.2 COMPARISON WITH EXISTING METHODS

**Quantitative Results.** Table 1 reports quantitative results for 2D-edits and 3D-edits. For **2D-edits**, GeoEdit leads on all seven metrics, including the lowest FID (25.07), DINOv2 distance (90.66), and Kernel Distance (0.051), and the highest Subject (0.910) and Background Consistency (0.977). It also achieves the lowest Warp Error (0.054) and Mean Distance (9.23), indicating strong realism and precise edits. For **3D-edits**, GeoEdit again surpasses prior methods with the lowest FID (64.30), DI-NOv2 distance (350.69), and Warp Error (0.051), while outperforming FreeFine (Zhu et al., 2025), GeoDiffuser (Sajnani et al., 2025), and DiffusionHandles (Pandey et al., 2024) on consistency met-

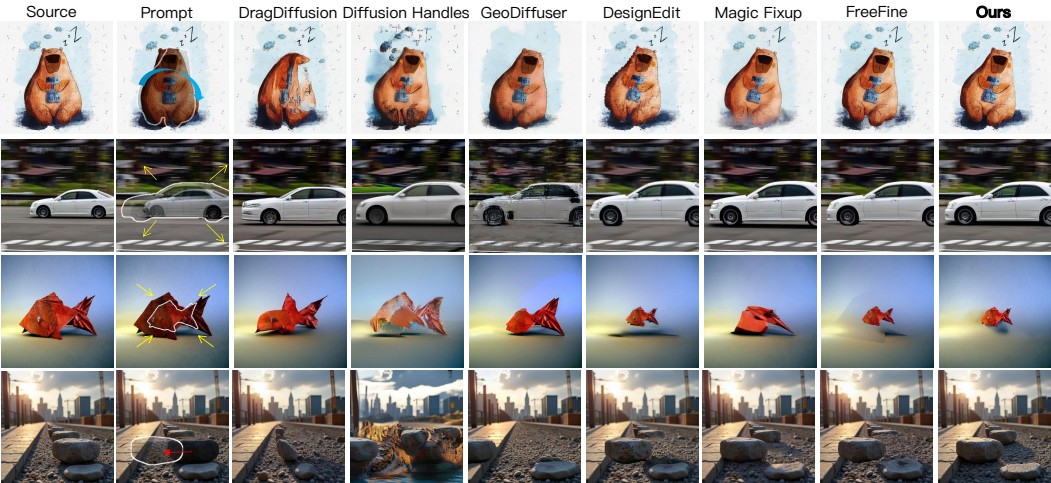

Figure 5: Qualitative comparison with different editing approaches on the 2D-edits of GeoBench.

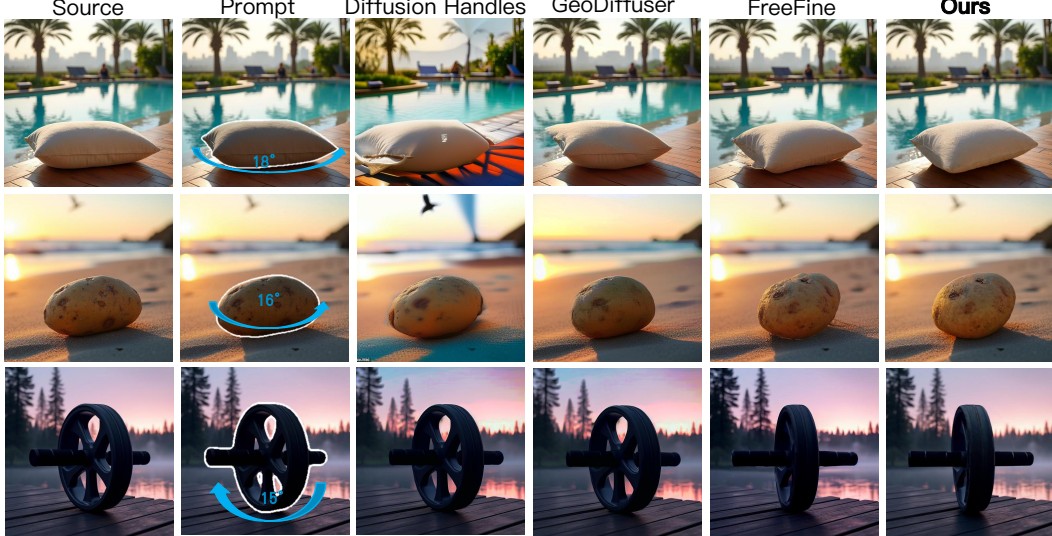

Figure 6: Qualitative comparison with different editing approaches on the 3D-edits of GeoBench.

rics. These results show that GeoEdit generalizes well across 2D and 3D transformations, producing high-quality, geometrically accurate edits.

**Qualitative Results.** In 2D-edits (Figure 5), such as resizing the goldfish, it preserves object structure and integrates coherently with the scene, avoiding artifacts seen in other methods. When removing the seal from the waterfront scene, our approach realistically reconstructs occluded regions and adjusts illumination to produce physically consistent shadows. For the more challenging 3D-edits (Figure 6), it is the only method producing perceptually convincing results, correctly reconstructing geometry and shading (e.g., the rotated wheel) where others fail. These results demonstrate precise, context-aware editing and improved realism for both 2D and 3D transformations.

## 5.3 USER STUDY

To provide a comprehensive subjective evaluation, we conducted a user study assessing the perceptual quality of **GeoEdit** across 2D-edits and 3D-edits tasks. Participants compared our method with competing approaches along three human-judged dimensions: *Quality*, *Consistency*, and *Effective-*

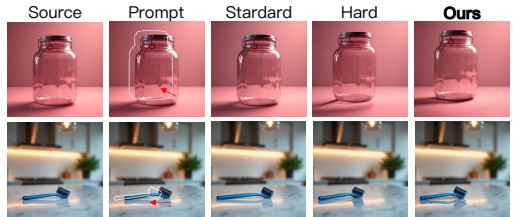

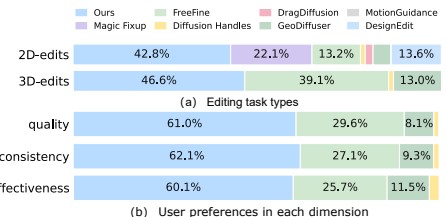

Figure 7: Ablation results on different attention modulation strategies.

Figure 8: User study results: GeoEdit outperforms prior methods on all criteria.

*ness*. As shown in Figure 8, **GeoEdit** consistently achieved the highest user-preference rates across all tasks and dimensions, substantially surpassing baseline methods. Further details—including annotator recruitment, sample construction, online survey interface, randomization to mitigate bias, and full per-dimension statistics are provided in Appendix E.

## 5.4 ABLATION STUDIES

Table 2: Ablation results of attention modulation and datasets. The best results are in bold.

| Setting | | Editing Tasks | Metrics | | | | | | |
|---|---|---|---|---|---|---|---|---|---|
| Type | Variant | | FID↓ | DINOv2↓ | KD↓ | SUBC↑ | BC↑ | WE↓ | MD↓ |
| Attention | Standard | 2D-edits | 29.11 | 115.04 | 0.052 | 0.891 | 0.969 | 0.097 | 15.75 |
| | Hard Modulation | | 27.09 | 107.83 | 0.051 | 0.899 | 0.964 | 0.063 | 11.11 |
| | Ours | | **25.28** | **94.79** | **0.051** | **0.908** | **0.977** | **0.057** | **9.32** |
| | Standard | 3D-edits | 68.02 | 372.68 | 0.055 | 0.823 | 0.962 | 0.082 | 22.50 |
| | Hard Modulation | | 67.32 | 367.27 | 0.055 | 0.832 | 0.969 | 0.061 | 19.62 |
| | Ours | | **64.30** | **350.69** | **0.054** | **0.840** | **0.977** | **0.051** | **18.08** |
| Dataset | Rendered | 2D-edits | 26.14 | 110.82 | 0.052 | 0.889 | 0.969 | 0.076 | 10.55 |
| | AIGC | | 25.82 | 106.03 | 0.051 | 0.898 | 0.972 | 0.066 | 9.96 |
| | Both | | **25.28** | **94.79** | **0.051** | **0.908** | **0.977** | **0.057** | **9.32** |
| | Rendered | 3D-edits | 66.92 | 385.10 | 0.055 | 0.822 | 0.961 | 0.091 | 19.04 |
| | AIGC | | 65.94 | 364.59 | 0.055 | 0.825 | 0.967 | 0.080 | 18.67 |
| | Both | | **64.30** | **350.69** | **0.054** | **0.840** | **0.977** | **0.051** | **18.08** |

We perform three ablation studies to quantify the contributions of data composition and attention modulation in GeoEdit.

**Data composition.** Table 2 shows that training on rendered data alone yields the weakest performance, while adding AIGC data improves all metrics. Combining rendered and AIGC data achieves the best scores (FID 25.28, DINOv2 94.79), indicating that diverse data strengthens geometric priors.

**Attention modulation.** Table 2 compares modulation strategies. Hard Modulation outperforms the baseline but sacrifices contextual cues, whereas our soft ESA attains the lowest FID (25.28) and Warp Error (0.057), with consistent gains on 3D-edits. According to Figure 7, ESA also enhances the model's ability to generate realistic lighting and shadow effects, further improving perceptual realism. Additional qualitative ablations are provided in Appendix F.

**Hyperparameter in ESA.** The hyperparameter $\alpha$ is set separately for object insertion and background restoration (Appendix G). Based on the results in Table 5, we use $\alpha_1 = 0.1$ and $\alpha_2 = 1$, which offer the best balance between edit fidelity and contextual consistency. Additional experiments under varied illumination (Appendix H) further validate these choices.

## 6    CONCLUSION

In this work, we introduced **GeoEdit**, a diffusion-based framework for geometric image editing that integrates a **Geometric Transformation** module, an in-context inpainting paradigm, and **Effects-Sensitive Attention** to jointly achieve precise object manipulation and realistic modeling of lighting and shadows. We also constructed a large-scale dataset of over 120,000 image pairs tailored for effective training, enabling **GeoEdit** to learn precise geometric transformations and realistic scene effects. As a result, extensive experiments and a user study demonstrate that **GeoEdit** consistently outperforms prior methods in geometric accuracy, visual fidelity, and realism.

## ACKNOWLEDGMENTS

This work was supported by the National Nature Science Foundation of China (Grant 62476029, 62225601, U23B2052), funded by the Fundamental Research Funds for the Beijing University of Posts and Telecommunications under Grant 2025TSQY08, and sponsored by Beijing Nova Program.

## ETHICS STATEMENT

Our research involves the creation and use of synthetic datasets for geometric image editing and associated human evaluations. All image data used in this work are either synthetically generated or obtained from publicly available sources with appropriate licenses. No personally identifiable information (PII) or sensitive content is included.

For the user study, participants were recruited from a professional annotation team. Participation was entirely voluntary, and all data were anonymized to protect privacy. The study protocol adhered to standard ethical guidelines, and no participants were exposed to harmful or distressing content.

Additionally, in manuscript preparation, any assistance provided by large language models (LLMs) was carefully reviewed and verified by the authors to ensure accuracy, avoid bias, and maintain scientific integrity.

Overall, we have taken care to ensure that our dataset, experimental protocols, and writing practices adhere to ethical standards appropriate for computer vision research.

## REPRODUCIBILITY STATEMENT

We have made extensive efforts to ensure the reproducibility of our work. The implementation details of our proposed method, including training configurations, model architectures, and hyperparameters, are provided in the main paper and further elaborated in the appendix. The source code package is included in the supplementary materials. For the datasets used in this work, we describe the preprocessing pipelines and filtering criteria in the main paper and appendix. For theoretical derivations, proofs, and assumptions are clearly documented in the appendix. Together, these resources are intended to enable researchers to fully reproduce our results.

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

# A  ATTENTION DISTRIBUTIONS AND DIVERGENCE ANALYSIS

This appendix complements Section 3.2 by formally analyzing how the standard attention, Hard Modulation, and ESA compares to an idealized "perfect" attention for our editing task. All notation follows Section 3.2.

## A.1  NECESSARY CONDITIONS FOR IDEAL ATTENTION

For geometric editing tasks, we argue that an ideal attention $A^\star$ should meet the following necessary conditions: (i) the attention weights are expected to be elevated in both object region and their effected region (shadows or reflections); (ii) Besides these regions, other non-salient parts require suppressed attention responses. For enhanced analytical precision, we decompose the auxiliary region into the effect region and the other region, i.e., $\mathcal{T}_{\text{aux}}^{(Q)} = \mathcal{T}_{\text{effect}}^{(Q)} \cup \mathcal{T}_{\text{other}}^{(Q)}$. Based on this, the necessary conditions for $A^\star$ can be formulated as:

$$
\begin{aligned}
&\forall_{q_i \in \mathcal{T}_{\text{obj}}^{(Q)}} : A_{ij}^\star \geq \rho; \\
&\forall_{q_i \in \mathcal{T}_{\text{effect}}^{(Q)}} : A_{ij}^\star \geq \rho; \\
&\forall_{q_i \in \mathcal{T}_{\text{obj}}^{(Q)} \cup \mathcal{T}_{\text{effect}}^{(Q)}} : \sum_{q_i} A_{ij}^\star \geq 1 - \varepsilon.
\end{aligned}
\tag{4}
$$

where $\rho \in (0,1)$ stands for the threshold for differentiating the critical region (i.e. $\mathcal{T}_{\text{obj}}^{(Q)} \cup \mathcal{T}_{\text{effect}}^{(Q)}$) from other non-critical region (i.e. $\mathcal{T}_{\text{other}}^{(Q)}$); $\varepsilon$ is a constant such that $\varepsilon \ll \rho$, indicating the non-critical region exhibits sufficiently small attention values.

## A.2  PROOF: STATEMENT (1) IN THEOREM 1

To prove $D_{\text{KL}}\big(A_{\cdot j}^\star \,\|\, A_{\cdot j}\big) > D_{\text{KL}}\big(A_{\cdot j}^\star \,\|\, A_{\cdot j}^{\text{ESA}}\big)$, it suffices to prove $D_{\text{KL}}\big(A_{\cdot j}^\star \,\|\, A_{\cdot j}\big) - D_{\text{KL}}\big(A_{\cdot j}^\star \,\|\, A_{\cdot j}^{\text{ESA}}\big) > 0$. To this end, we first come to the following equation:

$$
\begin{aligned}
&D_{\text{KL}}\big(A_{\cdot j}^\star \,\|\, A_{\cdot j}\big) - D_{\text{KL}}\big(A_{\cdot j}^\star \,\|\, A_{\cdot j}^{\text{ESA}}\big) \\
&= \left( \sum_{q_i \in \mathcal{T}_{\text{edit}}^{(Q)}} A_{ij}^\star \log \frac{A_{ij}^\star}{A_{ij}} + \sum_{q_i \in \mathcal{T}_{\text{aux}}^{(Q)}} A_{ij}^\star \log \frac{A_{ij}^\star}{A_{ij}} \right) - \left( \sum_{q_i \in \mathcal{T}_{\text{edit}}^{(Q)}} A_{ij}^\star \log \frac{A_{ij}^\star}{A_{ij}^{ESA}} + \sum_{q_i \in \mathcal{T}_{\text{aux}}^{(Q)}} A_{ij}^\star \log \frac{A_{ij}^\star}{A_{ij}^{ESA}} \right) \\
&= \underbrace{\left( \sum_{q_i \in \mathcal{T}_{\text{edit}}^{(Q)}} A_{ij}^\star \log \frac{A_{ij}^\star}{A_{ij}} - \sum_{q_i \in \mathcal{T}_{\text{edit}}^{(Q)}} A_{ij}^\star \log \frac{A_{ij}^\star}{A_{ij}^{ESA}} \right)}_{\Delta_{edit}} + \underbrace{\left( \sum_{q_i \in \mathcal{T}_{\text{aux}}^{(Q)}} A_{ij}^\star \log \frac{A_{ij}^\star}{A_{ij}} - \sum_{q_i \in \mathcal{T}_{\text{aux}}^{(Q)}} A_{ij}^\star \log \frac{A_{ij}^\star}{A_{ij}^{ESA}} \right)}_{\Delta_{aux}}.
\end{aligned}
\tag{5}
$$

Based on this, we can apply algebraic operations on $\Delta_{edit}$ and get that:

$$
\begin{aligned}
\Delta_{edit} &= \sum_{q_i \in \mathcal{T}_{\text{edit}}^{(Q)}} A_{ij}^\star \log \frac{A_{ij}^\star}{A_{ij}} - \sum_{q_i \in \mathcal{T}_{\text{edit}}^{(Q)}} A_{ij}^\star \log \frac{A_{ij}^\star}{A_{ij}^{ESA}} \\
&= \sum_{q_i \in \mathcal{T}_{\text{edit}}^{(Q)}} A_{ij}^\star \left( \log \frac{A_{ij}^\star}{A_{ij}} - \log \frac{A_{ij}^\star}{A_{ij}^{ESA}} \right) \\
&= \sum_{q_i \in \mathcal{T}_{\text{edit}}^{(Q)}} A_{ij}^\star \left( \log \frac{A_{ij}^\star}{A_{ij}} - \log \frac{A_{ij}^\star}{A_{ij}^{ESA}} \right) \\
&= \sum_{q_i \in \mathcal{T}_{\text{edit}}^{(Q)}} A_{ij}^\star \left( \log A_{ij}^{ESA} - \log A_{ij} \right).
\end{aligned}
\tag{6}
$$

Meanwhile, we can get the following equation by using the same operation as Eq.(6) on $\Delta_{aux}$:

$$
\Delta_{aux} = \sum_{q_i \in \mathcal{T}_{\text{aux}}^{(Q)}} A_{ij}^\star \left( \log A_{ij}^{ESA} - \log A_{ij} \right).
\tag{7}
$$

In this section, $\sum$ means sum over index $i$ by default. Then $\Delta_{edit}$ can be expressed as:

$$
\begin{aligned}
\Delta_{edit} &= \sum_{q_i \in \mathcal{T}_{\text{edit}}^{(Q)}} A_{ij}^\star \left( \log A_{ij}^{ESA} - \log A_{ij} \right) \\
&= \sum_{q_i \in \mathcal{T}_{\text{edit}}^{(Q)}} A_{ij}^\star \left( \log \frac{\exp(S_{ij}^{ESA})}{\sum \exp(S_{ij}^{ESA})} - \log \frac{\exp(S_{ij})}{\sum \exp(S_{ij})} \right) \\
&= \sum_{q_i \in \mathcal{T}_{\text{edit}}^{(Q)}} A_{ij}^\star \left( S_{ij}^{ESA} - S_{ij} + \log \sum \exp(S_{ij}) - \log \sum \exp(S_{ij}^{ESA}) \right) \\
&= \sum_{q_i \in \mathcal{T}_{\text{edit}}^{(Q)}} A_{ij}^\star \left[ \log \left( \frac{\exp(S_{ij}^{ESA})}{\exp(S_{ij})} \right) + \log \left( \frac{\sum \exp(S_{ij})}{\sum \exp(S_{ij}^{ESA})} \right) \right]
\end{aligned}
\tag{8}
$$

Then we analyze the term $\Delta_{aux}$. Note that we have $S_{ij}^{\text{ESA}} = S_{ij}$ when $q_i \in \mathcal{T}_{\text{aux}}^{(Q)}$. Thus, we can get that:

$$
\begin{aligned}
\Delta_{aux} &= \sum_{q_i \in \mathcal{T}_{\text{aux}}^{(Q)}} A_{ij}^\star \left( \log A_{ij}^{ESA} - \log A_{ij} \right) \\
&= \sum_{q_i \in \mathcal{T}_{\text{aux}}^{(Q)}} A_{ij}^\star \left( \log \frac{\exp(S_{ij}^{ESA})}{\sum \exp(S_{ij}^{ESA})} - \log \frac{\exp(S_{ij})}{\sum \exp(S_{ij})} \right) \\
&= \sum_{q_i \in \mathcal{T}_{\text{aux}}^{(Q)}} A_{ij}^\star \left( S_{ij}^{ESA} - S_{ij} + \log \sum \exp(S_{ij}) - \log \sum \exp(S_{ij}^{ESA}) \right) \\
&= \sum_{q_i \in \mathcal{T}_{\text{aux}}^{(Q)}} A_{ij}^\star \left( S_{ij} - S_{ij} + \log \sum \exp(S_{ij}) - \log \sum \exp(S_{ij}^{ESA}) \right) \\
&= \sum_{q_i \in \mathcal{T}_{\text{aux}}^{(Q)}} A_{ij}^\star \log \left( \frac{\sum \exp(S_{ij})}{\sum \exp(S_{ij}^{ESA})} \right).
\end{aligned}
\tag{9}
$$

Based on this, we can substitute Eq.(8) and Eq.(9) into Eq.(5), yielding that:

$$D_{\mathrm{KL}}\big(A^\star_{\cdot j} \,\|\, A_{\cdot j}\big) - D_{\mathrm{KL}}\big(A^\star_{\cdot j} \,\|\, A^{\mathrm{ESA}}_{\cdot j}\big)$$

$$= \sum_{q_i \in \mathcal{T}^{(Q)}_{\mathrm{edit}}} A^\star_{ij} \left[ \log\left(\frac{\exp(S^{ESA}_{ij})}{\exp(S_{ij})}\right) + \log\left(\frac{\sum \exp(S_{ij})}{\sum \exp(S^{ESA}_{ij})}\right) \right] + \sum_{q_i \in \mathcal{T}^{(Q)}_{\mathrm{aux}}} A^\star_{ij} \log\left(\frac{\sum \exp(S_{ij})}{\sum \exp(S^{ESA}_{ij})}\right)$$

$$= \sum_{q_i \in \mathcal{T}^{(Q)}_{\mathrm{edit}}} A^\star_{ij} \log\left(\frac{\exp(S^{ESA}_{ij})}{\exp(S_{ij})}\right) + \left[ \sum_{q_i \in \mathcal{T}^{(Q)}_{\mathrm{edit}}} A^\star_{ij} \log\left(\frac{\sum \exp(S_{ij})}{\sum \exp(S^{ESA}_{ij})}\right) + \sum_{q_i \in \mathcal{T}^{(Q)}_{\mathrm{aux}}} A^\star_{ij} \log\left(\frac{\sum \exp(S_{ij})}{\sum \exp(S^{ESA}_{ij})}\right) \right]$$

$$= \sum_{q_i \in \mathcal{T}^{(Q)}_{\mathrm{edit}}} A^\star_{ij} \log\left(\frac{\exp(S^{ESA}_{ij})}{\exp(S_{ij})}\right) + \sum_{q_i \in \mathcal{T}^{(Q)}} A^\star_{ij} \log\left(\frac{\sum \exp(S_{ij})}{\sum \exp(S^{ESA}_{ij})}\right)$$

$$= \sum_{q_i \in \mathcal{T}^{(Q)}_{\mathrm{edit}}} A^\star_{ij} \log\left(\frac{\exp(S^{ESA}_{ij})}{\exp(S_{ij})}\right) + \log\left(\frac{\sum \exp(S_{ij})}{\sum \exp(S^{ESA}_{ij})}\right)$$

$$\geq \sum_{q_i \in \mathcal{T}^{(Q)}_{\mathrm{edit}}} \rho \log\left(\frac{\exp(S^{ESA}_{ij})}{\exp(S_{ij})}\right) + \log\left(\frac{\sum \exp(S_{ij})}{\sum \exp(S^{ESA}_{ij})}\right).$$

$$(10)$$

Furthermore, we can derive that:

$$\log\left(\frac{\exp(S^{ESA}_{ij})}{\exp(S_{ij})}\right) = \log\left(\frac{\exp(S_{ij} + \delta)}{\exp(S_{ij})}\right) = \log\left(\frac{\exp(S_{ij})\exp(\delta)}{\exp(S_{ij})}\right) = \delta. \tag{11}$$

Therefore, the first term in the last line of Eq.(10) can be reformulated as:

$$\sum_{q_i \in \mathcal{T}^{(Q)}_{\mathrm{edit}}} \rho \log\left(\frac{\exp(S^{ESA}_{ij})}{\exp(S_{ij})}\right) = \sum_{q_i \in \mathcal{T}^{(Q)}_{\mathrm{edit}}} \rho \cdot \delta = |\mathcal{T}^{(Q)}_{\mathrm{edit}}| \cdot \rho \cdot \delta. \tag{12}$$

Now we come to the second term in the last line of Eq.(10):

$$\log\left(\frac{\sum \exp(S_{ij})}{\sum \exp(S^{ESA}_{ij})}\right)$$

$$= \log\left(\frac{\sum \exp(S_{ij})}{\sum \exp(S_{ij} + \delta)}\right) \tag{13}$$

$$= \log\left(\frac{\sum \exp(S_{ij})}{\sum \exp(S_{ij})\exp(\delta)}\right) = \log\left(\frac{1}{\exp(\delta)}\right) = -\delta.$$

By substituting Eq.(12) and Eq.(13) into Eq.(10), we could know that:

$$D_{\mathrm{KL}}\big(A^\star_{\cdot j} \,\|\, A_{\cdot j}\big) - D_{\mathrm{KL}}\big(A^\star_{\cdot j} \,\|\, A^{\mathrm{ESA}}_{\cdot j}\big) \geq |\mathcal{T}^{(Q)}_{\mathrm{edit}}| \cdot \rho \cdot \delta - \delta = \underbrace{\delta(|\mathcal{T}^{(Q)}_{\mathrm{edit}}| \cdot \rho - 1)}_{\Delta_3}. \tag{14}$$

In this way, when $\rho \geq 1/|\mathcal{T}^{(Q)}_{\mathrm{edit}}|$, we have $\Delta_3 \geq 0$, indicating that $D_{\mathrm{KL}}\big(A^\star_{\cdot j} \,\|\, A_{\cdot j}\big) \geq D_{\mathrm{KL}}\big(A^\star_{\cdot j} \,\|\, A^{\mathrm{ESA}}_{\cdot j}\big)$. Now the proof is concluded.

## A.3 PROOF: STATEMENT (2) IN THEOREM 1

Based on Eq.(14), if we want to prove $D_{\mathrm{KL}}\big(A^\star_{\cdot j} \,\|\, A^{\mathrm{hard}}_{\cdot j}\big) > D_{\mathrm{KL}}\big(A^\star_{\cdot j} \,\|\, A^{ESA}_{\cdot j}\big)$, we only need to prove $D_{\mathrm{KL}}\big(A^\star_{\cdot j} \,\|\, A^{\mathrm{hard}}_{\cdot j}\big) > D_{\mathrm{KL}}\big(A^\star_{\cdot j} \,\|\, A_{\cdot j}\big)$. To prove $D_{\mathrm{KL}}\big(A^\star_{\cdot j} \,\|\, A^{\mathrm{hard}}_{\cdot j}\big) > D_{\mathrm{KL}}\big(A^\star_{\cdot j} \,\|\, A_{\cdot j}\big)$, it suffices to establish the existence of a finite upper bound for $D_{\mathrm{KL}}\big(A^\star_{\cdot j} \,\|\, A^{\mathrm{ESA}}_{\cdot j}\big)$. To this end, we first analyze the structure of the standard attention distribution 1 .

$$A_{ij} = g(S_{ij}), \quad S_{ij} = \frac{q_i k_j^\top}{\sqrt{d}}.$$

Since all queries $q_i$ and keys $k_j$ are finite-valued vectors, the logits $S_{ij} = \frac{q_i k_j^\top}{\sqrt{d}}$ are finite for all $i, j$. Consequently, $A_{ij} > 0$ for all $i, j$, ensuring that the support of $A_{\cdot j}$ covers the entire query token set $\mathcal{T}^{(Q)}$.

Now, consider the ideal distribution $A_{\cdot j}^\star$ which satisfies Eq. 4 with $p_{\text{aux}} > 0$. Since $A_{ij} > 0$ for all $i$, the ratio $A_{ij}^\star / A_{ij}$ is finite for every $i$, and the KL divergence:

$$D_{\text{KL}}\big(A_{\cdot j}^\star \,\|\, A_{\cdot j}\big) = \sum_{q_i} A_{ij}^\star \log \frac{A_{ij}^\star}{A_{ij}} \tag{15}$$

is a finite weighted sum of finite terms. More precisely, we can bound it as follows:

Let $\beta = \min_{i,j} A_{ij} > 0$ (which exists since there are finitely many tokens and $A_{ij} > 0$). Then for any $i$ with $A_{ij}^\star > 0$, we have:

$$\log \frac{A_{ij}^\star}{A_{ij}} \leq \log \frac{1}{A_{ij}} \leq \log \frac{1}{\beta}.$$

Similarly, since $A_{ij}^\star \leq 1$, we have:

$$A_{ij}^\star \log \frac{A_{ij}^\star}{A_{ij}} \leq \log \frac{1}{\beta}.$$

Therefore,

$$D_{\text{KL}}\big(A_{\cdot j}^\star \,\|\, A_{\cdot j}\big) \leq |\mathcal{T}^{(Q)}| \cdot \log \frac{1}{\beta} < +\infty. \tag{16}$$

In contrast, as shown in Section A.4, when $p_{\text{aux}} > 0$, we have:

$$D_{\text{KL}}\big(A_{\cdot j}^\star \,\|\, A_{\cdot j}^{\text{hard}}\big) = +\infty.$$

Thus, we conclude that:

$$D_{\text{KL}}\big(A_{\cdot j}^\star \,\|\, A_{\cdot j}^{\text{hard}}\big) = +\infty > D_{\text{KL}}\big(A_{\cdot j}^\star \,\|\, A_{\cdot j}\big), \tag{17}$$

By integrating Eq.(14) and Eq.(17), the proof can be concluded.

## A.4 KL DIVERGENCE BETWEEN THE HARD MODULATION AND IDEAL DISTRIBUTIONS

Recall the Hard Modulation logits and the resulting attention Eq. 2 :

$$A_{ij}^{\text{hard}} = g\big(S_{ij}^{\text{hard}}\big), \qquad S_{ij}^{\text{hard}} = \begin{cases} +\infty, & q_i \in \mathcal{T}_{\text{edit}}^{(Q)}, \\ \dfrac{q_i k_j^\top}{\sqrt{d}}, & q_i \in \mathcal{T}_{\text{aux}}^{(Q)}, \end{cases}$$

Applying the softmax over $i$, the limit of $A_{ij}^{\text{hard}}$ is

$$A_{ij}^{\text{hard}} = \begin{cases} 1/|\mathcal{T}_{\text{edit}}^{(Q)}|, & q_i \in \mathcal{T}_{\text{edit}}^{(Q)}, \\ 0, & q_i \in \mathcal{T}_{\text{aux}}^{(Q)}, \end{cases} \tag{18}$$

The KL divergence between the ideal and Hard Modulation distributions for each $j$ is

$$D_{\text{KL}}\big(A_{\cdot j}^\star \,\|\, A_{\cdot j}^{\text{hard}}\big) = \sum_{q_i} A_{ij}^\star \log \frac{A_{ij}^\star}{A_{ij}^{\text{hard}}}. \tag{19}$$

Because $A_{ij}^{\text{hard}} = 0$ for all $q_i \in \mathcal{T}_{\text{aux}}^{(Q)}$, any nonzero ideal mass $A_{ij}^\star > 0$ in the auxiliary region yields

$$A_{ij}^\star \log \frac{A_{ij}^\star}{0} = +\infty. \tag{20}$$

Thus, whenever the ideal distribution places even a small positive mass $p_{\text{aux}} > 0$ on auxiliary tokens (as allowed by Eq. 4), we have

$$D_{\text{KL}}\big(A_{\cdot j}^\star \,\|\, A_{\cdot j}^{\text{hard}}\big) = +\infty. \tag{21}$$

**Interpretation.** Hard Modulation collapses all probability mass onto the edit region and assigns zero probability to auxiliary tokens. Since the ideal distribution generally preserves at least a small positive mass on the auxiliary region, the two distributions have disjoint support, and their KL divergence diverges. This highlights the instability of overly hard attention gating compared to a softer, ideal allocation.

## B  VISUAL COMPARISON: RENDERED DATASET VS. OTHER RENDERING DATASETS

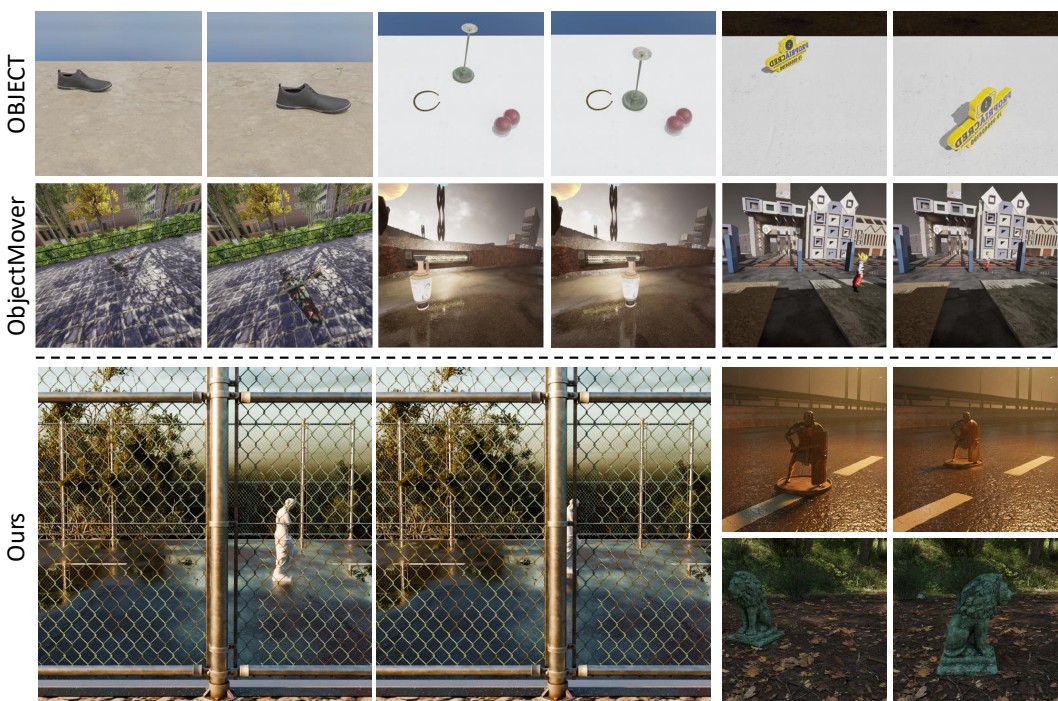

Figure 9: Visual comparison between our Render Dataset and existing rendering datasets, highlighting its broader scene diversity, richer geometric variations, and higher visual fidelity.

We compare our Rendered Dataset with the datasets rendered by ObjectMover (Yu et al., 2025) and 3DiT (Michel et al., 2023). As shown in Figure 9, our dataset provides more realistic scenes and objects than the 3DiT dataset, and also captures more complex lighting and shadow phenomena. Compared to the ObjectMover dataset, our Rendered Dataset includes not only more diverse and realistic environments but also more challenging object–scene interactions (e.g., in the lower-left example of Figure 9, a moving sculpture is partially occluded by a wire fence). Our Rendered Dataset also captures detailed object geometry and appearance, making it one of the most comprehensive and high-quality datasets for object movement currently available.

## C  DATA FILTERING CRITERIA

To ensure high-quality and physically plausible samples, we conducted a large-scale human-in-the-loop filtering process over three weeks with a 20-person annotation team. Each image–mask pair was manually inspected according to the following criteria:

**Spatial coherence.** Verify the spatial coherence of the moved object's placement. The object must be integrated naturally within its environment, demonstrating a believable relationship with surrounding objects in terms of scale, contact, and alignment. This requires ensuring it rests securely on supporting surfaces without floating, maintaining appropriate clearance to avoid interpenetration,

and following the scene's functional logic to prevent unrealistic positioning. The overall goal is to achieve a seamless and contextually appropriate fit.

**Feature consistency.** It is imperative to ensure that the manipulation of an object preserves the absolute integrity of its intrinsic visual properties. This means the object's fundamental characteristics—including its geometric form (shape, scale), surface qualities (texture, material), and color values must remain invariant relative to its original state. Any dynamic changes, such as perspective adjustments, should be applied consistently across the entire scene to maintain this fidelity.

**Illumination consistency.** Ensure the moved object's lighting and shadows are physically correct. Verify that shadow direction, softness, and length match the scene's lights, and that it correctly occludes light and casts contact shadows to avoid a floating look.

**Controlled generation.** Confirm that the edited image strictly adheres to the intended transformation instructions without introducing unexpected artifacts or deviations in content.

Only samples satisfying all four criteria were retained, resulting in over 100,000 high-quality image–mask pairs in the final AIGC Dataset.

## D  IMPLEMENTATION DETAILS

Our **GeoEdit** builds upon FLUX.1 Fill [dev] (Black Forest Labs, 2024), an inpainting model based on the DiT architecture. We replaced the original T5 text encoder (Raffel et al., 2020) with a SigLIP image encoder (Zhai et al., 2023) for textual inputs, fine-tuned via LoRA (rank 256) (Hu et al., 2022). All images were processed at a resolution of $1024\times1024$ with a batch size of 4. Training used the Prodigy optimizer (Mishchenko & Defazio, 2023) with safeguard warmup, bias correction, and 0.01 weight decay, on 4 NVIDIA H100 GPUs (80GB each). The model was trained on the proposed dataset for about 5000 steps. Inference employed 28 denoising iterations, and the training objective followed the flow matching framework (Lipman et al., 2022).

## E  USER STUDY DETAILS

To comprehensively evaluate perceptual quality, we conducted a formal user study. The study was structured around three key dimensions: (1) *Quality*, assessing visual realism and the absence of artifacts; (2) *Consistency*, evaluating the preservation of the original subject and background integrity; and (3) *Effectiveness*, measuring how accurately the result realizes the intended geometric transformation.

The study was hosted as an online survey to ensure reproducibility. We recruited 33 professional annotators to evaluate 40 distinct instances randomly sampled from GeoBench (Zhu et al., 2025) (10 instances each for Move, Rotate, Resize, and 3D-edits). In each trial, participants viewed the source image, a visualization of the editing prompt, and the anonymized outputs from all competing methods. To mitigate positional bias, both the sample presentation and the method outputs were fully randomized. For each instance, annotators selected the top-1 result that best satisfied each of the three perceptual criteria, yielding a total of 3,960 valid votes.

The baseline methods varied by task. For 2D-edits, we compared against DesignEdit (Jia et al., 2024), Diffusion Handles (Pandey et al., 2024), DragDiffusion (Shi et al., 2024), FreeFine (Zhu et al., 2025), GeoDiffuser (Sajnani et al., 2025), Magic Fixup (Alzayer et al., 2025), and MotionGuidance (Geng & Owens, 2024). For 3D-edits, the comparison set included Diffusion Handles, FreeFine, and GeoDiffuser.

As detailed in Fig. 8 and Table 3, **GeoEdit** consistently achieved the highest user preference rates across both 2D and 3D edits. Furthermore, when aggregated by perceptual dimension, our method secured a decisive lead in Quality, Consistency, and Effectiveness. This strong subjective preference, which aligns with our quantitative findings, underscores the practical utility and robustness of **GeoEdit** in producing high-fidelity, artifact-free results that faithfully execute user commands.

Table 3: User research results statistics. (Votes aggregated across three perceptual dimensions: Image Quality, Consistency, and Editing Effectiveness.)

| Method | 2D-Edits | | | | 3D-Edits | Total |
|---|---|---|---|---|---|---|
| | Move | Resize | Rotate | 2D Total | | |
| DesignEdit(Jia et al., 2024) | 207 | 159 | 38 | 404 | | 404 |
| DragDiffusion(Shi et al., 2024) | 8 | 29 | 25 | 62 | | 62 |
| Magic Fixup(Alzayer et al., 2025) | 213 | 169 | 274 | 656 | | 656 |
| MotionGuidance (Geng & Owens, 2024) | 0 | 1 | 0 | 1 | | 1 |
| Diffusion Handles(Pandey et al., 2024) | 15 | 16 | 8 | 39 | 13 | 52 |
| FreeFine (Zhu et al., 2025) | 88 | 93 | 210 | 391 | 387 | 778 |
| GeoDiffuser(Sajnani et al., 2025) | 75 | 29 | 42 | 146 | 129 | 275 |
| **Ours** | **384** | **494** | **393** | **1271** | **461** | **1732** |

## F  ADDITIONAL VISUALIZATION OF ATTENTION MODULATION ABLATION RESULTS

In figure 10, we present additional visualizations of different attention-modulation variants. We observe that ESA produces lighting and shadow effects that are more realistic and closer to the ground truth, demonstrating its superior performance.

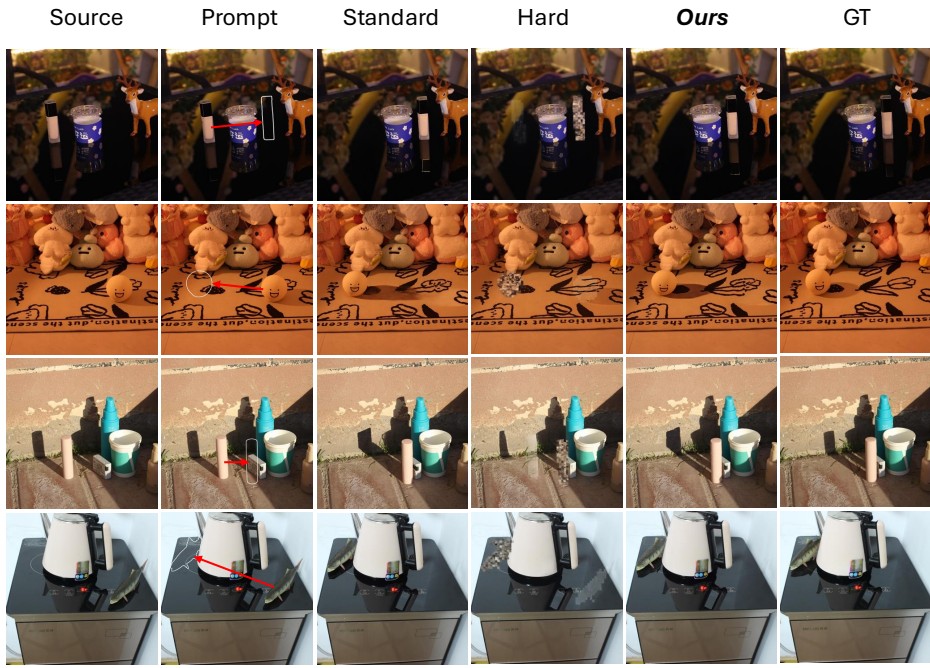

Figure 10: Qualitative visualizations of attention modulation ablation results.

## G  ABLATION STUDY ON THE HYPERPARAMETERS IN ESA

As discussed in Section 3, geometric editing involves both object insertion and background restoration. Accordingly, we introduce two hyperparameters, $\alpha_1$ and $\alpha_2$, which control the scaling of attention in the insertion and restoration regions. In the table 5, We explore the effects of different

settings for these hyperparameters, and the results indicate that $\alpha_1 = 0.1$ and $\alpha_2 = 1$ yield the best overall performance.

Table 4: Ablation Study on the Hyperparameters in ESA and the best results are in bold.

| Hyperparameters | | Editing Tasks | Metrics | | | | | | |
|---|---|---|---|---|---|---|---|---|---|
| $\alpha_1$ | $\alpha_2$ | | FID↓ | DINOv2↓ | KD↓ | SUBC↑ | BC↑ | WE↓ | MD↓ |
| 1.0 | 0.1 | | 27.09 | 107.75 | 0.051 | 0.900 | 0.978 | 0.063 | 11.33 |
| 1.0 | 0.5 | | 28.24 | 111.74 | 0.051 | 0.900 | 0.977 | 0.061 | 10.57 |
| 1.0 | 1.0 | | 27.05 | 105.01 | 0.051 | 0.898 | 0.976 | 0.071 | 11.03 |
| 0.5 | 0.1 | | 27.13 | 107.97 | 0.051 | 0.898 | 0.977 | 0.068 | 13.26 |
| 0.5 | 0.5 | 2D-edits | 28.68 | 112.00 | 0.051 | 0.896 | 0.977 | 0.071 | 10.40 |
| 0.5 | 1.0 | | 29.11 | 114.77 | 0.051 | 0.894 | 0.975 | 0.075 | 10.95 |
| 0.1 | 0.1 | | 26.07 | 99.54 | 0.051 | 0.897 | **0.978** | 0.074 | 13.79 |
| 0.1 | 0.5 | | 28.08 | 112.52 | 0.051 | 0.897 | 0.977 | 0.072 | 10.70 |
| 0.1 | 1.0 | | **25.28** | **94.79** | **0.051** | **0.908** | 0.977 | **0.057** | **9.32** |
| 1.0 | 0.1 | | 71.35 | 387.55 | 0.055 | 0.827 | 0.980 | 0.086 | 20.07 |
| 1.0 | 0.5 | | 73.35 | 399.68 | 0.055 | 0.823 | 0.978 | 0.066 | 21.22 |
| 1.0 | 1.0 | | 69.56 | 381.68 | 0.055 | 0.831 | 0.978 | 0.089 | 18.98 |
| 0.5 | 0.1 | | 69.75 | 385.01 | 0.055 | 0.829 | 0.977 | 0.078 | 19.82 |
| 0.5 | 0.5 | 3D-edits | 67.90 | 355.50 | 0.055 | 0.833 | 0.979 | 0.099 | 18.14 |
| 0.5 | 1.0 | | 72.56 | 394.12 | 0.055 | 0.828 | 0.973 | 0.090 | 20.47 |
| 0.1 | 0.1 | | 67.96 | 360.67 | 0.055 | 0.833 | **0.980** | 0.092 | 19.01 |
| 0.1 | 0.5 | | 69.57 | 367.44 | 0.055 | 0.832 | 0.979 | 0.097 | 19.99 |
| 0.1 | 1.0 | | **64.30** | **350.69** | **0.054** | **0.840** | 0.977 | **0.051** | **18.08** |

# H  ABLATION STUDY OF ESA HYPERPARAMETERS UNDER DIFFERENT ILLUMINATION CONDITIONS

Table 5: Ablation Study on the Hyperparameters in ESA under High-illumination Setting. Best results are in bold.

| Hyperparameters | | Editing Tasks | Metrics | | | | | | |
|---|---|---|---|---|---|---|---|---|---|
| $\alpha_1$ | $\alpha_2$ | | FID↓ | DINOv2↓ | KD↓ | SUBC↑ | BC↑ | WE↓ | MD↓ |
| 1.0 | 0.1 | | 27.19 | 107.87 | 0.051 | 0.901 | 0.977 | 0.064 | 11.28 |
| 1.0 | 0.5 | | 28.33 | 111.85 | 0.051 | 0.898 | 0.979 | 0.059 | 10.52 |
| 1.0 | 1.0 | | 27.14 | 105.13 | 0.051 | 0.897 | 0.975 | 0.073 | 11.00 |
| 0.5 | 0.1 | | 27.25 | 108.09 | 0.051 | 0.897 | 0.978 | 0.067 | 13.22 |
| 0.5 | 0.5 | 2D-edits | 28.79 | 112.12 | 0.051 | 0.895 | 0.976 | 0.072 | 10.36 |
| 0.5 | 1.0 | | 29.21 | 114.89 | 0.051 | 0.893 | 0.974 | 0.076 | 10.91 |
| 0.1 | 0.1 | | 26.17 | 99.66 | 0.051 | 0.898 | **0.979** | 0.073 | 13.75 |
| 0.1 | 0.5 | | 28.18 | 112.64 | 0.051 | 0.896 | 0.978 | 0.071 | 10.66 |
| 0.1 | 1.0 | | **25.37** | **94.89** | **0.051** | **0.907** | 0.976 | **0.058** | **9.28** |
| 1.0 | 0.1 | | 71.42 | 387.61 | 0.055 | 0.826 | 0.981 | 0.085 | 20.09 |
| 1.0 | 0.5 | | 73.28 | 399.58 | 0.055 | 0.824 | 0.979 | 0.067 | 21.18 |
| 1.0 | 1.0 | | 69.51 | 381.73 | 0.055 | 0.830 | 0.977 | 0.088 | 18.96 |
| 0.5 | 0.1 | | 69.71 | 384.95 | 0.055 | 0.830 | 0.976 | 0.079 | 19.80 |
| 0.5 | 0.5 | 3D-edits | 67.86 | 355.44 | 0.055 | 0.832 | **0.980** | 0.098 | 18.12 |
| 0.5 | 1.0 | | 72.60 | 394.18 | 0.055 | 0.829 | 0.974 | 0.091 | 20.49 |
| 0.1 | 0.1 | | 67.93 | 360.71 | 0.055 | 0.832 | 0.981 | 0.093 | 19.03 |
| 0.1 | 0.5 | | 69.54 | 367.38 | 0.055 | 0.833 | 0.978 | 0.096 | 19.97 |
| 0.1 | 1.0 | | **64.27** | **350.63** | **0.054** | **0.839** | 0.976 | **0.050** | **18.06** |

To evaluate whether ESA's improvement is sensitive to the choice of the scaling factor $\alpha_1$ and $\alpha_2$ under different lighting conditions, we extended our ablation study by categorizing the evaluation data into three illumination regimes: high, medium, and low light. We then conducted separate

Table 6: Ablation Study on the Hyperparameters in ESA under Medium-illumination Setting. Best results are in bold.

| Hyperparameters | | Editing Tasks | Metrics | | | | | | |
|---|---|---|---|---|---|---|---|---|---|
| $\alpha_1$ | $\alpha_2$ | | FID↓ | DINOv2↓ | KD↓ | SUBC↑ | BC↑ | WE↓ | MD↓ |
| 1.0 | 0.1 | | 27.01 | 107.65 | 0.051 | 0.899 | 0.978 | 0.062 | 11.37 |
| 1.0 | 0.5 | | 28.16 | 111.63 | 0.051 | 0.901 | 0.976 | 0.063 | 10.61 |
| 1.0 | 1.0 | | 27.00 | 104.91 | 0.051 | 0.899 | 0.977 | 0.069 | 11.07 |
| 0.5 | 0.1 | | 27.04 | 107.93 | 0.051 | 0.899 | 0.976 | 0.069 | 13.30 |
| 0.5 | 0.5 | 2D-edits | 28.59 | 111.96 | 0.051 | 0.897 | 0.978 | 0.070 | 10.44 |
| 0.5 | 1.0 | | 29.02 | 114.73 | 0.052 | 0.895 | 0.974 | 0.074 | 10.99 |
| 0.1 | 0.1 | | 26.00 | 99.42 | 0.051 | 0.896 | 0.977 | 0.075 | 13.83 |
| 0.1 | 0.5 | | 28.01 | 112.40 | 0.051 | 0.898 | 0.976 | 0.073 | 10.74 |
| 0.1 | 1.0 | | **25.21** | **94.69** | **0.051** | **0.909** | **0.977** | **0.056** | **9.36** |
| 1.0 | 0.1 | | 71.31 | 387.47 | 0.055 | 0.828 | 0.979 | 0.087 | 20.04 |
| 1.0 | 0.5 | | 73.41 | 399.80 | 0.055 | 0.822 | 0.977 | 0.065 | 21.26 |
| 1.0 | 1.0 | | 69.62 | 381.62 | 0.055 | 0.832 | 0.979 | 0.090 | 19.00 |
| 0.5 | 0.1 | | 69.80 | 385.09 | 0.055 | 0.828 | 0.978 | 0.077 | 19.84 |
| 0.5 | 0.5 | 3D-edits | 67.95 | 355.59 | 0.055 | 0.834 | 0.978 | 0.100 | 18.16 |
| 0.5 | 1.0 | | 72.53 | 394.06 | 0.055 | 0.827 | 0.972 | 0.089 | 20.44 |
| 0.1 | 0.1 | | 68.00 | 360.59 | 0.055 | 0.834 | 0.979 | 0.091 | 19.00 |
| 0.1 | 0.5 | | 69.61 | 367.50 | 0.055 | 0.831 | **0.980** | 0.098 | 20.02 |
| 0.1 | 1.0 | | **64.34** | **350.75** | **0.055** | **0.841** | 0.978 | **0.052** | **18.10** |

Table 7: Ablation Study on the Hyperparameters in ESA under Low-illumination Setting. Best results are in bold.

| Hyperparameters | | Editing Tasks | Metrics | | | | | | |
|---|---|---|---|---|---|---|---|---|---|
| $\alpha_1$ | $\alpha_2$ | | FID↓ | DINOv2↓ | KD↓ | SUBC↑ | BC↑ | WE↓ | MD↓ |
| 1.0 | 0.1 | | 27.08 | 107.73 | 0.051 | 0.900 | 0.977 | 0.063 | 11.31 |
| 1.0 | 0.5 | | 28.23 | 111.72 | 0.051 | 0.900 | 0.977 | 0.061 | 10.57 |
| 1.0 | 1.0 | | 27.04 | 105.00 | 0.051 | 0.898 | 0.976 | 0.071 | 11.03 |
| 0.5 | 0.1 | | 27.12 | 107.95 | 0.051 | 0.898 | 0.977 | 0.068 | 13.26 |
| 0.5 | 0.5 | 2D-edits | 28.67 | 111.98 | 0.051 | 0.896 | 0.977 | 0.071 | 10.40 |
| 0.5 | 1.0 | | 29.10 | 114.75 | 0.051 | 0.894 | 0.975 | 0.075 | 10.95 |
| 0.1 | 0.1 | | 26.06 | 99.54 | 0.051 | 0.897 | 0.978 | 0.074 | 13.79 |
| 0.1 | 0.5 | | 28.07 | 112.52 | 0.051 | 0.897 | 0.977 | 0.072 | 10.70 |
| 0.1 | 1.0 | | **25.27** | **94.79** | **0.051** | **0.908** | **0.977** | **0.057** | **9.32** |
| 1.0 | 0.1 | | 71.29 | 387.51 | 0.055 | 0.827 | 0.980 | 0.086 | 20.08 |
| 1.0 | 0.5 | | 73.39 | 399.75 | 0.055 | 0.823 | 0.979 | 0.065 | 21.24 |
| 1.0 | 1.0 | | 69.60 | 381.65 | 0.055 | 0.831 | 0.978 | 0.089 | 18.99 |
| 0.5 | 0.1 | | 69.78 | 385.02 | 0.055 | 0.829 | 0.977 | 0.078 | 19.83 |
| 0.5 | 0.5 | 3D-edits | 67.93 | 355.52 | 0.055 | 0.833 | 0.979 | 0.099 | 18.15 |
| 0.5 | 1.0 | | 72.55 | 394.09 | 0.055 | 0.828 | 0.973 | 0.090 | 20.46 |
| 0.1 | 0.1 | | 67.98 | 360.64 | 0.055 | 0.833 | **0.980** | 0.092 | 19.01 |
| 0.1 | 0.5 | | 69.59 | 367.45 | 0.055 | 0.832 | 0.979 | 0.097 | 20.00 |
| 0.1 | 1.0 | | **64.32** | **350.70** | **0.054** | **0.840** | 0.977 | **0.051** | **18.09** |

experiments for each group using the same set of $\alpha$ values. As shown in tables 5 6 7, the $\alpha_1$ and $\alpha_2$ selected in our main experiments consistently yields the best performance across all three lighting conditions. This indicates that ESA's gains are stable and not strongly dependent on illumination-specific tuning.

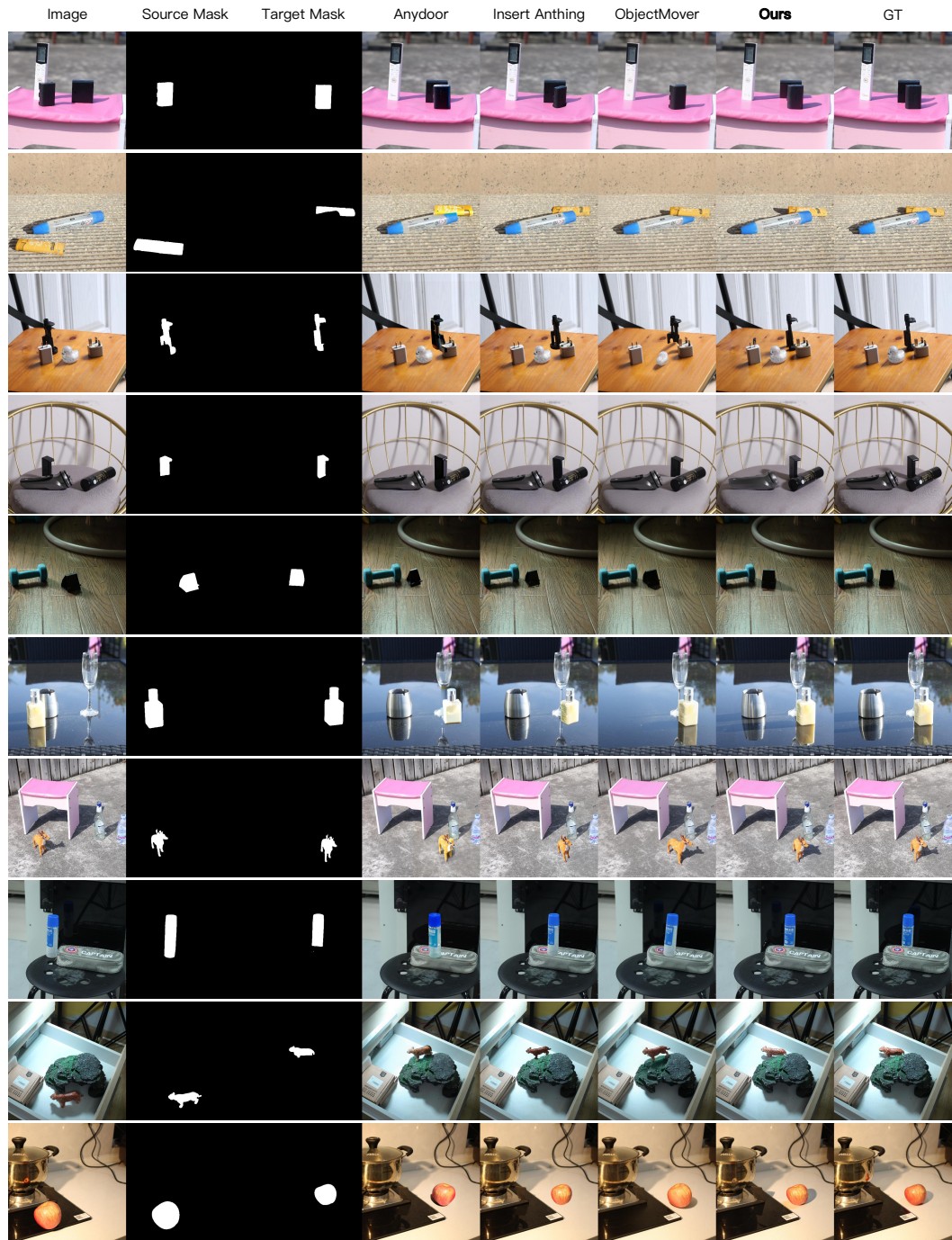

Figure 11: Qualitative comparison of different editing approaches on the ObjMove-A dataset.

# I    EXPERIMENTAL EVALUATION ON THE OBJMOVE-A (YU ET AL., 2025) BENCHMARK

We conduct a comprehensive evaluation on ObjMove-A, a benchmark dataset providing ground-truth images for object movement tasks. We employ a suite of eight metrics to quantify different aspects of performance. To evaluate the fidelity of the inserted object, we compute object-level metrics which include DINO-Score (Caron et al., 2021), CLIP-Score (Radford et al., 2021), and

Table 8: Quantitative results on object insertion and manipulation tasks. We report eight metrics for image quality, consistency, and editing effectiveness. Our method achieves the best performance.

| Methods | PSNR↑ | SSIM↑ | Clip-Score↑ | DINO-Score↑ | FID↓ | Lpips↓ | ReMOVE↑ | DreamSim↓ |
|---|---|---|---|---|---|---|---|---|
| 3DiT Michel et al. (2023) | 18.18 | 0.817 | 57.41 | 57.41 | 95.65 | 0.461 | 0.651 | 0.164 |
| AnyDoor Chen et al. (2024) | 23.42 | 0.897 | 85.90 | 85.90 | 41.11 | 0.212 | 0.860 | 0.059 |
| DreamFuse Huang et al. (2025a) | 18.55 | 0.815 | 82.04 | 82.04 | 76.04 | 0.326 | 0.480 | 0.106 |
| Insert Anything Song et al. (2025) | 22.30 | 0.842 | 86.21 | 86.21 | 44.79 | 0.297 | 0.848 | 0.055 |
| Magic Fixup Alzayer et al. (2025) | 22.83 | 0.842 | 80.51 | 80.51 | 45.09 | 0.330 | 0.844 | 0.055 |
| ObjectMover Yu et al. (2025) | 24.06 | 0.856 | 82.18 | 82.18 | 36.84 | 0.327 | 0.874 | 0.045 |
| **Ours** | **26.18** | **0.891** | **92.90** | **92.90** | **23.65** | **0.180** | **0.874** | **0.023** |

DreamSim (Fu et al., 2023) on the cropped target region. Since the task also requires removing the object from its original location, we use ReMOVE (Chandrasekar et al., 2024). We further measure overall image similarity using PSNR and SSIM applied to the full generated image. Additionally, we report FID (Heusel et al., 2017) and LPIPS (Zhang et al., 2018) to evaluate global realism and perceptual quality.

Among the baselines, AnyDoor (Chen et al., 2024), DreamFuse (Huang et al., 2025a), and Insert Anything (Song et al., 2025) are object insertion models. In our pipeline, we first use OmniEraser (Wei et al., 2025) to remove the original object and obtain a clean background, after which the object is inserted into the target location. Quantitative results demonstrate that our approach achieves state-of-the-art performance across the majority of these metrics, a superiority that is further evidenced by the visual comparisons presented in Figure 11, where our model produces the most realistic and geometrically consistent edits among all existing methods.

## J    Limitations and Future Work

GeoEdit may struggle in scenes with strong secondary physical effects. For example, when a motorcycle kicks up sand or dust, moving the object may not fully transfer these extended effects to the new location. These dynamic, particle-based phenomena are fundamentally different from illumination-related effects (e.g., shading or soft shadows), which GeoEdit is specifically designed to handle. In future work, we plan to further reduce the model's reliance on external 3D priors, improve its ability to model complex physical interactions, and strengthen its handling of illumination and shadow generation.

## K    The Use of Large Language Models (LLMs)

During the preparation of this manuscript, we leveraged large language models (LLMs) to assist with several aspects of writing and document organization. First, LLMs were used to search for relevant literature based on our research focus and to generate corresponding BibTeX entries for cited papers. Second, they assisted in formatting and refining tables, ensuring consistent style, alignment, and readability across the manuscript. All outputs suggested or generated by LLMs were carefully reviewed and edited by the authors to maintain correctness, clarity, and adherence to the intended scientific meaning. This workflow allowed us to improve efficiency in manuscript preparation while retaining full authorial control over the content.

## L    Source Code Availability

The source code package is provided in the supplementary materials to facilitate reproducibility and enable readers to experiment with our proposed methods.

