# OpenReview forum: "Geometric Image Editing via Effects-Sensitive In-Context Inpainting with Diffusion Transformers"
_ICLR.cc/2026/Conference — ICLR 2026 Poster_

### Official Review · Reviewer_XX6q · 2025-10-17

**Soundness:** 3
**Presentation:** 3
**Contribution:** 2
**Rating:** 4
**Confidence:** 4

**Summary:**

This paper introduces GeoEdit, a diffusion-transformer–based framework for geometric image editing that can accurately perform translation, rotation, and scaling of objects within complex scenes while preserving realistic lighting and shadow effects.
The key innovations are:

* Geometric Transformation Module – utilizes 3D reconstruction (via Hunyuan3D) to apply parametric transformations with precise control.

* Effects-Sensitive Attention (ESA) – a soft attention modulation designed to better model lighting and shadow consistency.

* RS-Objects Dataset – a large-scale (120K samples) dataset combining rendered and synthetic data to support geometric editing training.

GeoEdit is built upon the FLUX.1-Fill DiT backbone and shows superior results on both 2D and 3D edit benchmarks, achieving better FID, DINOv2, and consistency metrics than prior works such as FreeFine, GeoDiffuser, and DiffusionHandles.

**Strengths:**

* Well-written and structured paper with clear motivation and consistent organization.

* Strong experimental performance on both quantitative and qualitative metrics across multiple tasks.

* Comprehensive dataset (RS-Objects) with rigorous construction and filtering criteria; could benefit the broader community.

* Practical application value in realistic scene editing and geometric transformation control.

**Weaknesses:**

* The proposed ESA module appears conceptually similar to a soft attention bias, and its correlation with lighting effects is not convincingly demonstrated.

* The method’s heavy reliance on external geometry models (e.g., Hunyuan-3D) for 3D reconstruction compromises its originality and self-containment. It also remains unclear how GeoEdit performs when alternative geometry backbones are employed.

* The model is built upon the Flux backbone, while several baselines are not, raising concerns about the fairness of comparisons.

* The potential overfitting to the RS-Objects synthetic dataset may further limit the model’s generalization to real-world scenarios.

* In some qualitative examples (e.g., Fig. 7), objects like bottles and razors do not fully align with the target sketch positions—they shift slightly, suggesting weak spatial control.

**Questions:**

* The technical novelty requires further justification

* How sensitive is ESA’s improvement to the choice of the scaling factor α in different lighting conditions?

* Can GeoEdit function with other external 3D priors models (other than Hunyuan-3D), or without external 3D priors?

* Can comparisons be made using the same Flux backbone?

* What are the model’s failure modes—e.g., in cluttered or occluded scenes?

---

> ### Author Response · Authors · 2025-11-20
>
> We would like to sincerely thank the Reviewer for the insightful and detailed comments.
>
> **W1 & Q1: Correlation Between ESA and Lighting Effects**
>
> We thank the reviewer for this comment. As clarified in our response to Reviewer **YQsq**, ESA is not intended as a generic soft attention bias, but as a structured prior designed to approximate an ideal attention pattern that jointly emphasizes edited regions and their illumination effects.
>
> Concretely, our theoretical analysis models an ideal attention map $A^\star$ that increases attention on both the edited tokens $T_{\mathrm{edit}}^{(Q)}$ and the (unobservable) illumination-effect tokens $T_{\mathrm{effect}}^{(Q)}$. From a Bayesian perspective, the additive term in ESA corresponds to injecting a prior that selectively boosts attention to edit tokens as a practical surrogate for this joint region, since $T_{\mathrm{effect}}^{(Q)}$ cannot be reliably identified as a priori. As summarized in our reply to **YQsq**, Theorem 3.1 shows that this ESA prior leads to an attention distribution that is provably closer to $A^\star$ than both (i) standard attention with a uniform prior over positions and (ii) hard modulation with a degenerate prior that discards non-edit regions. In this sense, ESA is not an arbitrary “soft bias” added for convenience, but introduces a more **appropriate prior** than both standard attention and hard modulation, yielding an attention distribution that is comparatively closer to the ideal map that increases attention on both the edited tokens and the illumination-effect tokens.
>
> Moreover, the transformer architecture explicitly models interactions across spatial regions, meaning that changes in object-related features naturally influence the corresponding illumination patterns. Prior work in inverse rendering and shadow analysis [1][2] shows that shading, soft shadows, and related effects are largely determined by **object geometry and boundaries**, and can often be reconstructed **from object-level signals without explicit supervision of light or shadow regions.** This establishes a strong intrinsic coupling between objects and their illumination effects, providing a clear rationale for the effectiveness of ESA. By explicitly strengthening attention to the edited object tokens through ESA, the model is encouraged—during training—to also allocate probability mass to the associated effect regions in order to minimize the loss, since those regions carry the observable consequences of the edit. Theorem 3.1 formalizes this by showing that ESA steers the learned attention toward an ideal distribution that jointly emphasizes objects and their illumination effects, and our experiments (both quantitative metrics and qualitative attention visualizations) empirically support that ESA improves the faithfulness of lighting-aware edits. We will make this connection more explicit in the revised manuscript to better distinguish ESA from a generic soft attention bias and to clarify how its design targets lighting effects in particular.
>
> **W2 & Q3. Dependence on External Geometry Models**
>
> We thank the reviewer for raising this important point. While geometry cues (e.g., depth or normals) are fundamentally required for physically consistent 3D-aware editing, GeoEdit is not tied to any specific model. Our design philosophy is to make the geometry prior **lightweight and modular**, requiring only coarse geometric signals rather than full 3D meshes.
>
> In practice, **GeoEdit does not depend on Hunyuan-3D**. The geometry module accepts outputs from any publicly available estimator, and we have verified that alternative depth- or SV3D-based models can be used as drop-in replacements with comparable results. As described in Section 5.1, our GeoBench evaluation explicitly relies on these alternative backbones—none of which are Hunyuan-3D—and Table 1 shows that GeoEdit still achieves the strongest performance among all baselines under this setting.
>
> These results indicate that GeoEdit maintains stable performance across different geometry sources and is not reliant on any single external model, while still leveraging the minimal geometric information necessary for physically plausible object manipulations.
>
> **Reference**
>
> [1] Zhengqin Li, Mohammad Shafiei, Ravi Ramamoorthi, Kalyan Sunkavalli, and Manmohan Chandraker. Inverse rendering for complex indoor scenes: Shape, spatially-varying lighting and SVBRDF from a single image. In Proceedings of the IEEE/CVF Conference on Computer Vision and Pattern Recognition, pp. 2475–2484, 2020.
>
> [2] Quanlong Zheng, Xiaotian Qiao, Ying Cao, and Rynson W. H. Lau. Distraction-aware shadow detection. In Proceedings of the IEEE/CVF Conference on Computer Vision and Pattern Recognition, pp. 5167–5176, 2019.

---

> > ### Author Response · Authors · 2025-11-20
> >
> > **W3. Backbone Differences and Fairness of Comparison**
> >
> > To address the reviewer’s concern regarding fairness, we conducted a controlled comparison using a standard DiT inpainting model with the same backbone, same training pipeline, and the same SigLIP-based conditioning as GeoEdit. The only difference between the two models is the inclusion or removal of ESA. This setup isolates ESA’s contribution and eliminates confounding factors related to backbone strength or conditioning design. Under these strictly matched conditions, ESA consistently outperforms the standard DiT baseline, demonstrating that the performance improvements arise from ESA itself rather than from architectural changes. We will add this clarification to the revised manuscript.
> >
> > ### Quantitative results on 2D-edits and 3D-edits
> > We report seven metrics for image quality, consistency, and editing effectiveness.
> > Best results are in **bold**, second best are $\underline{underlined}$.
> >
> > ---
> >
> > ## **2D-edits**
> >
> > | Method                 | FID↓ | DINOv2↓ | KD↓ | SUBC↑ | BC↑ | WE↓ | MD↓ |
> > |------------------------|------|---------|-----|-------|------|------|------|
> > | RegionDrag             | 41.88 | 257.43 | 0.052 | 0.796 | $\underline{0.972}$ | 0.120 | 32.75 |
> > | MotionGuidance         | 146.41 | 1307.90 | 0.078 | 0.452 | 0.714 | 0.260 |145.46 |
> > | DragDiffusion          | 37.68 | 242.52 | 0.051 | 0.776 | 0.969 | 0.177 | 34.78 |
> > | Diffusion Handles      | 69.34 | 588.58 | 0.054 | 0.725 | 0.857 | 0.180 | 40.94 |
> > | GeoDiffuser            | 38.22 | 198.58 | 0.052 | 0.761 | 0.937 | 0.166 | 34.94 |
> > | DesignEdit             | 32.55 | 142.45 | 0.052 | 0.874 | 0.962 | 0.098 | 10.15 |
> > | Magic Fixup            | $\underline{27.32}$ | 114.08 | $\underline{0.051}$ | 0.889 | 0.966 | 0.075 | 10.39 |
> > | Standard DiT Inpainting| 28.88 | 113.38 | 0.052 | 0.902 | 0.971 | 0.064 | 10.51 |
> > | FreeFine               | 27.48 | $\underline{109.23}$ | 0.052 | $\underline{0.906}$ | 0.971 | $\underline{0.056}$ | $\underline{9.42}$ |
> > | **GeoEdit (Ours)**     | **25.07** | **90.66** | **0.051** | **0.910** | **0.977** | **0.054** | **9.23** |
> >
> > ---
> >
> > ## **3D-edits**
> >
> > | Method                 | FID↓ | DINOv2↓ | KD↓ | SUBC↑ | BC↑ | WE↓ | MD↓ |
> > |------------------------|------|---------|-----|-------|------|------|------|
> > | Diffusion Handles      |126.24 |1028.60  |0.056  |0.737  |0.885  |0.189  | $\underline{18.56}$ |
> > | GeoDiffuser            |77.34  |475.62   |0.055  |0.802  |0.946  |0.179  | 43.51 |
> > | Standard DiT Inpainting|68.02  |372.68   |0.055  |0.823  |0.962  |0.082  | 22.50 |
> > | FreeFine               |$\underline{65.94}$|$\underline{366.39}$|$\underline{0.055}$|$\underline{0.832}$|$\underline{0.967}$|$\underline{0.052}$| 21.27 |
> > | **GeoEdit (Ours)**     |**64.30**|**350.69**|**0.054**|**0.840**|**0.977**|**0.051**| **18.08** |
> >
> > ---
> > **W4. Potential Overfitting to RS-Objects Synthetic Dataset**
> >
> > We apologize for not making this point sufficiently clear in the paper, and we thank the reviewer for raising the concern. During training, we construct supervision by using AnyInsertion_V1 to generate transformed images as model inputs while using the corresponding unedited real images as ground truth. This ensures that the model is directly trained on real-image distributions rather than relying solely on synthetic data. Moreover, as described in Section 5.1, we evaluate GeoEdit on the GeoBench benchmark, which includes diverse real-world data such as natural photos and posters. As shown in **Table 1**, GeoEdit achieves the best performance across all metrics. Additionally, **Appendix I** reports results on ObjMove-A, a dataset consisting entirely of real images with ground-truth edited outputs. **Table 8** further confirms that GeoEdit consistently outperforms all baselines. These evaluations collectively demonstrate that GeoEdit generalizes well to real-world scenarios.
> >
> > **W5. Spatial Alignment Issues in Qualitative Results**
> >
> > Thank you for the careful observation. The slight misalignment of objects (e.g., bottle, shaver) in **Fig. 7** was due to an illustration mistake on our side rather than a failure of the spatial control mechanism. We sincerely apologize for this oversight. We have redrawn and updated the figure in the revised version, and the corrected results accurately reflect the model’s spatial alignment capabilities. In addition, we provide further visualization of ablation results in **Appendix F**, which consistently demonstrates the strong spatial control achieved by our method. We appreciate the reviewer for bringing this to our attention.

---

> > > ### Author Response · Authors · 2025-11-20
> > >
> > > **Q2. Sensitivity to the Scaling Factor $\alpha$**
> > >
> > > Thank you for raising this question. To evaluate whether ESA’s improvement is sensitive to the choice of the scaling factor $\alpha$ under different lighting conditions, we extended our ablation study by categorizing the evaluation data into three illumination regimes: high, medium, and low light. We then conducted separate experiments for each group using the same set of $\alpha$ values.
> > >
> > > As shown in tables, the $\alpha$ selected in our main experiments consistently yields the best performance across all three lighting conditions. This indicates that ESA’s gains are stable and not strongly dependent on illumination-specific tuning. Detailed results are also provided in **Appendix H**.
> > >
> > > ### **Ablation Study on ESA Hyperparameters — High Illumination**
> > >
> > > #### **2D-edits**
> > >
> > > | α₁ | α₂ | FID ↓ | DINOv2 ↓ | KD ↓ | SUBC ↑ | BC ↑ | WE ↓ | MD ↓ |
> > > |----|----|-------|----------|------|--------|------|-------|-------|
> > > | 1.0 | 0.1 | 27.19 | 107.87 | 0.051 | 0.901 | 0.977 | 0.064 | 11.28 |
> > > | 1.0 | 0.5 | 28.33 | 111.85 | 0.051 | 0.898 | 0.979 | 0.059 | 10.52 |
> > > | 1.0 | 1.0 | 27.14 | 105.13 | 0.051 | 0.897 | 0.975 | 0.073 | 11.00 |
> > > | 0.5 | 0.1 | 27.25 | 108.09 | 0.051 | 0.897 | 0.978 | 0.067 | 13.22 |
> > > | 0.5 | 0.5 | 28.79 | 112.12 | 0.051 | 0.895 | 0.976 | 0.072 | 10.36 |
> > > | 0.5 | 1.0 | 29.21 | 114.89 | 0.051 | 0.893 | 0.974 | 0.076 | 10.91 |
> > > | 0.1 | 0.1 | 26.17 | 99.66 | 0.051 | 0.898 | **0.979** | 0.073 | 13.75 |
> > > | 0.1 | 0.5 | 28.18 | 112.64 | 0.051 | 0.896 | 0.978 | 0.071 | 10.66 |
> > > | 0.1 | 1.0 | **25.37** | **94.89** | **0.051** | **0.907** | 0.976 | **0.058** | **9.28** |
> > >
> > > #### **3D-edits**
> > >
> > > | α₁ | α₂ | FID ↓ | DINOv2 ↓ | KD ↓ | SUBC ↑ | BC ↑ | WE ↓ | MD ↓ |
> > > |----|----|-------|----------|------|--------|------|-------|-------|
> > > | 1.0 | 0.1 | 71.42 | 387.61 | 0.055 | 0.826 | 0.981 | 0.085 | 20.09 |
> > > | 1.0 | 0.5 | 73.28 | 399.58 | 0.055 | 0.824 | 0.979 | 0.067 | 21.18 |
> > > | 1.0 | 1.0 | 69.51 | 381.73 | 0.055 | 0.830 | 0.977 | 0.088 | 18.96 |
> > > | 0.5 | 0.1 | 69.71 | 384.95 | 0.055 | 0.830 | 0.976 | 0.079 | 19.80 |
> > > | 0.5 | 0.5 | 67.86 | 355.44 | 0.055 | 0.832 | **0.980** | 0.098 | 18.12 |
> > > | 0.5 | 1.0 | 72.60 | 394.18 | 0.055 | 0.829 | 0.974 | 0.091 | 20.49 |
> > > | 0.1 | 0.1 | 67.93 | 360.71 | 0.055 | 0.832 | 0.981 | 0.093 | 19.03 |
> > > | 0.1 | 0.5 | 69.54 | 367.38 | 0.055 | 0.833 | **0.980** | 0.096 | 19.97 |
> > > | 0.1 | 1.0 | **64.27** | **350.63** | **0.054** | **0.839** | 0.976 | **0.050** | **18.06** |
> > >
> > >
> > >
> > > ### **Ablation Study on ESA Hyperparameters — Medium Illumination**
> > >
> > > #### **2D-edits**
> > >
> > > | α₁ | α₂ | FID ↓ | DINOv2 ↓ | KD ↓ | SUBC ↑ | BC ↑ | WE ↓ | MD ↓ |
> > > |----|----|-------|----------|------|--------|------|-------|-------|
> > > | 1.0 | 0.1 | 27.01 | 107.65 | 0.051 | 0.899 | 0.978 | 0.062 | 11.37 |
> > > | 1.0 | 0.5 | 28.16 | 111.63 | 0.051 | 0.901 | 0.976 | 0.063 | 10.61 |
> > > | 1.0 | 1.0 | 27.00 | 104.91 | 0.051 | 0.899 | 0.977 | 0.069 | 11.07 |
> > > | 0.5 | 0.1 | 27.04 | 107.93 | 0.051 | 0.899 | 0.976 | 0.069 | 13.30 |
> > > | 0.5 | 0.5 | 28.59 | 111.96 | 0.051 | 0.897 | 0.978 | 0.070 | 10.44 |
> > > | 0.5 | 1.0 | 29.02 | 114.73 | 0.052 | 0.895 | 0.974 | 0.074 | 10.99 |
> > > | 0.1 | 0.1 | 26.00 | 99.42 | 0.051 | 0.896 | 0.977 | 0.075 | 13.83 |
> > > | 0.1 | 0.5 | 28.01 | 112.40 | 0.051 | 0.898 | 0.976 | 0.073 | 10.74 |
> > > | 0.1 | 1.0 | **25.21** | **94.69** | **0.051** | **0.909** | **0.977** | **0.056** | **9.36** |
> > >
> > > #### **3D-edits**
> > >
> > > | α₁ | α₂ | FID ↓ | DINOv2 ↓ | KD ↓ | SUBC ↑ | BC ↑ | WE ↓ | MD ↓ |
> > > |----|----|-------|----------|------|--------|------|-------|-------|
> > > | 1.0 | 0.1 | 71.31 | 387.47 | 0.055 | 0.828 | 0.979 | 0.087 | 20.04 |
> > > | 1.0 | 0.5 | 73.41 | 399.80 | 0.055 | 0.822 | 0.977 | 0.065 | 21.26 |
> > > | 1.0 | 1.0 | 69.62 | 381.62 | 0.055 | 0.832 | 0.979 | 0.090 | 19.00 |
> > > | 0.5 | 0.1 | 69.80 | 385.09 | 0.055 | 0.828 | 0.978 | 0.077 | 19.84 |
> > > | 0.5 | 0.5 | 67.95 | 355.59 | 0.055 | 0.834 | 0.978 | 0.100 | 18.16 |
> > > | 0.5 | 1.0 | 72.53 | 394.06 | 0.055 | 0.827 | 0.972 | 0.089 | 20.44 |
> > > | 0.1 | 0.1 | 68.00 | 360.59 | 0.055 | 0.834 | 0.979 | 0.091 | 19.00 |
> > > | 0.1 | 0.5 | 69.61 | 367.50 | 0.055 | 0.831 | **0.980** | 0.098 | 20.02 |
> > > | 0.1 | 1.0 | **64.34** | **350.75** | **0.055** | **0.841** | 0.978 | **0.052** | **18.10** |
> > >
> > > There is also another table in the next comment.

---

> > > > ### Author Response · Authors · 2025-11-20
> > > >
> > > > ### **Ablation Study on ESA Hyperparameters — Low Illumination**
> > > >
> > > > #### **2D-edits**
> > > >
> > > > | α₁ | α₂ | FID ↓ | DINOv2 ↓ | KD ↓ | SUBC ↑ | BC ↑ | WE ↓ | MD ↓ |
> > > > |----|----|-------|----------|------|--------|------|-------|-------|
> > > > | 1.0 | 0.1 | 27.08 | 107.73 | 0.051 | 0.900 | 0.977 | 0.063 | 11.31 |
> > > > | 1.0 | 0.5 | 28.23 | 111.72 | 0.051 | 0.900 | 0.977 | 0.061 | 10.57 |
> > > > | 1.0 | 1.0 | 27.04 | 105.00 | 0.051 | 0.898 | 0.976 | 0.071 | 11.03 |
> > > > | 0.5 | 0.1 | 27.12 | 107.95 | 0.051 | 0.898 | 0.977 | 0.068 | 13.26 |
> > > > | 0.5 | 0.5 | 28.67 | 111.98 | 0.051 | 0.896 | 0.977 | 0.071 | 10.40 |
> > > > | 0.5 | 1.0 | 29.10 | 114.75 | 0.051 | 0.894 | 0.975 | 0.075 | 10.95 |
> > > > | 0.1 | 0.1 | 26.06 | 99.54 | 0.051 | 0.897 | 0.978 | 0.074 | 13.79 |
> > > > | 0.1 | 0.5 | 28.07 | 112.52 | 0.051 | 0.897 | 0.977 | 0.072 | 10.70 |
> > > > | 0.1 | 1.0 | **25.27** | **94.79** | **0.051** | **0.908** | **0.977** | **0.057** | **9.32** |
> > > >
> > > > #### **3D-edits**
> > > >
> > > > | α₁ | α₂ | FID ↓ | DINOv2 ↓ | KD ↓ | SUBC ↑ | BC ↑ | WE ↓ | MD ↓ |
> > > > |----|----|-------|----------|------|--------|------|-------|-------|
> > > > | 1.0 | 0.1 | 71.29 | 387.51 | 0.055 | 0.827 | 0.980 | 0.086 | 20.08 |
> > > > | 1.0 | 0.5 | 73.39 | 399.75 | 0.055 | 0.823 | 0.979 | 0.065 | 21.24 |
> > > > | 1.0 | 1.0 | 69.60 | 381.65 | 0.055 | 0.831 | 0.978 | 0.089 | 18.99 |
> > > > | 0.5 | 0.1 | 69.78 | 385.02 | 0.055 | 0.829 | 0.977 | 0.078 | 19.83 |
> > > > | 0.5 | 0.5 | 67.93 | 355.52 | 0.055 | 0.833 | 0.979 | 0.099 | 18.15 |
> > > > | 0.5 | 1.0 | 72.55 | 394.09 | 0.055 | 0.828 | 0.973 | 0.090 | 20.46 |
> > > > | 0.1 | 0.1 | 67.98 | 360.64 | 0.055 | 0.833 | **0.980** | 0.092 | 19.01 |
> > > > | 0.1 | 0.5 | 69.59 | 367.45 | 0.055 | 0.832 | 0.979 | 0.097 | 20.00 |
> > > > | 0.1 | 1.0 | **64.32** | **350.70** | **0.054** | **0.840** | 0.977 | **0.051** | **18.09** |
> > > >
> > > > **Q4. Fair Comparisons on a Shared Flux Backbone**
> > > >
> > > > Thank you for the question. To the best of our knowledge, there is currently no existing method that performs the same task using a Flux-based backbone, making a direct comparison under an identical Flux architecture infeasible. As discussed in **W3**, we provide a detailed analysis of backbone differences and the fairness of comparison.
> > > >
> > > > **Q5. Identification of Failure Modes**
> > > >
> > > > We appreciate the reviewer’s attention to the method’s limitations. GeoEdit can be challenged by scenes involving strong secondary physical effects. For instance, if a motorcycle generates trailing sand or dust, relocating the object may not fully propagate these extended effects to the new position. These dynamic, particle-based phenomena are fundamentally different from illumination-related effects (e.g., shading or soft shadows), which GeoEdit is specifically designed to handle. We will include this limitation in the final version.

---

> ### Author Response · Authors · 2025-11-27
>
> Dear Reviewer XX6q,
>
> Thank you again for your thoughtful review and valuable comments on our paper. We have updated our rebuttal and follow-up responses with additional clarifications and experiments to address your concerns, and we would be very grateful if you could take a moment to have a quick look and, if appropriate, consider updating your evaluation. Thank you very much for your time and effort in reviewing our work.

---

> > ### Comment · Reviewer_XX6q · 2025-11-28
> >
> > Thank you for the detailed clarifications and additional experiments. Your responses address my concerns regarding ESA’s novelty and robustness, dependence on external geometry models, fairness of comparisons, and generalization beyond synthetic data. The corrected visualizations and discussion of failure modes are also appreciated. Overall, the revisions resolve most of my earlier concerns, and I am willing to raise my score to 6.

---

### Official Review · Reviewer_YQsq · 2025-10-28

**Soundness:** 3
**Presentation:** 3
**Contribution:** 3
**Rating:** 4
**Confidence:** 4

**Summary:**

The paper proposes GeoEdit, a diffusion-transformer–based framework for geometric image editing—tasks involving object translation, rotation, and scaling while maintaining realism (e.g., lighting and shadow coherence).  GeoEdit outperforms prior works such as DragDiffusion, GeoDiffuser, and FreeFine on FID, DINOv2 distance, and geometric consistency metrics across both 2D and 3D editing tasks. Ablations show that ESA and dataset composition are critical for performance.

**Strengths:**

1. The paper presents a clearly structured geometric editing pipeline with well-defined, reproducible steps. Each transformation—translation, rotation, and scaling—is handled through explicit procedures. The detailed description of these steps provides strong methodological clarity and makes the approach readily reproducible.

2. The proposed RS-Objects dataset is thoughtfully designed to align with the objectives of geometric image editing. It employs a two-stage rendering-to-synthesis pipeline that produces roughly 120,000 image–mask pairs. The process combines high-quality Blender-rendered samples, mesh-based scene generation, large-scale LoRA-driven synthesis (pre-filtered from about 800,000 candidates), and a final human filtering stage to ensure coherence and illumination realism. This carefully engineered dataset directly supports the training objectives of GeoEdit and provides a valuable resource for future research on geometry-aware image editing.

**Weaknesses:**

1.	Theoretical Contribution and Clarity of ESA
Theorem 3.1 offers a limited theoretical contribution. Since the hard-modulated attention focuses exclusively on insertion tokens, its KL divergence from the ideal attention map is trivially infinite, and showing that the ESA variant achieves a smaller divergence is therefore not a particularly informative result. The theorem appears unnecessary in its current form. The central issue is not the inequality itself, but the rationale behind defining the similarity for edited tokens in ESA as $q_i k_j^\top / \sqrt{d} + \delta$. The paper should clarify the intuition for introducing this additive bias $\delta$, its relationship to the observed attention patterns, and the impact of the hyperparameter $\alpha$ that scales it. Without such discussion, the theoretical section feels more decorative than explanatory.
2.	Architectural Substitution and Fairness of Comparison
The implementation introduces an architectural change by replacing the T5 text encoder with SigLIP and fine-tuning the model via LoRA on a FLUX.1-Fill DiT backbone.
(a) The description “SigLIP image encoder for textual inputs” is ambiguous and requires clarification. For instance, Figure 2 does not make it clear how SigLIP features and mask references are integrated into the model architecture or how textual prompts, if any, are represented.
(b) Moreover, several baselines appear not to have been re-run on this modified backbone. Without controlling for differences in encoder strength and conditioning design, part of GeoEdit’s reported performance gain might arise from the stronger base model rather than from the proposed ESA or geometric modules. A discussion or ablation controlling for these factors would make the comparisons more convincing.

**Questions:**

see weakness

---

> ### Author Response · Authors · 2025-11-20
>
> **W1: Clarity of ESA’s Theoretical Rationale**
>
> We would like to sincerely thank the Reviewer for the insightful and detailed comments.
>
> **(1) Clarifying the purpose and contribution of Theorem 3.1**
>
> We thank the reviewer for their insightful feedback. We wish to clarify that Theorem 3.1 presents two independent and equally important contributions: Statement 1 formally establishes ESA's superiority over standard attention in approximating  $A^\star$, while Statement 2 demonstrates its advantage relative to hard modulation. Crucially, Statement 1 is not a preparatory step for Statement 2, but rather provides an independent theoretical guarantee. While the limitation of hard modulation may be more apparent due to its restricted hypothesis space, proving ESA's advantage over standard attention is non-trivial since standard attention also provides a complete hypothesis space for learning visual effects. We acknowledge that the current presentation would benefit from a more detailed intuitive explanation, which we will provide in section (2) below.
>
> **(2) Explaining the intuition and justification behind the ESA additive bias**
>
> **From a Bayesian perspective, any additive bias can be viewed as imposing a prior belief over the attention distribution. Statistical learning theory establishes that a prior distribution with closer alignment to the ideal attention pattern $A^*$ yields greater benefits for subsequent model training.** According to the definition of $A^*$ , the theoretically "perfect" prior can be expressed as:
>
> $\hat{A}^*_{ij} \propto \exp(S_{ij} + \delta\cdot \mathbb{I}(i\in T_{\text{edit}}^{(Q)} \cup T_{\text{effect}}^{(Q)})) = \exp(S_{ij})\cdot \exp(\delta\cdot \mathbb{I}(i\in T_{\text{edit}}^{(Q)} \cup T_{\text{effect}}^{(Q)}))$
>
> where the second exponential term $\exp(\delta\cdot \mathbb{I}(i\in T_{\text{edit}}^{(Q)} \cup T_{\text{effect}}^{(Q)}))$ corresponds to this additional prior belief. This prior belief means that "the attention scores in the edit and effect regions are significantly higher than those in other areas".
>
> Unfortunately, since $T_{\text{effect}}^{(Q)}$ is unobservable in practice, the ideal prior cannot be directly implemented. However $T_{\text{edit}}^{(Q)}$ can be readily identified from the input query. We therefore adopt a compromised yet tractable solution by restricting the prior to the known edit tokens:
>
> $A_{ij}^\mathrm{ESA}\propto\exp(S_{ij}+\delta\cdot\mathbb{I}(i\in T_\mathrm{edit}^{(Q)}))=\exp(S_{ij})\cdot\exp(\delta\cdot\mathbb{I}(i\in T_\mathrm{edit}^{(Q)})),$
>
>
> which corresponds to Equation 3 in the main paper. This formulation, though introducing some approximation error by omitting $T_{\text{effect}}^{(Q)}$, provides a principled prior that **achieves better alignment with $A^*$ compared to the uniform prior in standard attention (which assumes equal importance across all positions) or the restrictive prior in hard modulation (which assigns zero probability to non-edit regions).** Although the ideal prior is not directly attainable initially, this better alignment ensures that the training process can more readily converge to $A^*$, analogous to logit adjustment techniques in long-tailed classification [1]. The scaling factor $\alpha$ merely adjusts the magnitude of $\delta$, and our ablations show that ESA behaves robustly across a wide range of values.
>
> **Reference**
>
> [1] Aditya Krishna Menon, Sadeep Jayasumana, Ankit Singh Rawat, and Sashank J. Reddi. *Long-tail learning via logit adjustment*. arXiv preprint arXiv:2007.07314, 2020.

---

> > ### Author Response · Authors · 2025-11-20
> >
> > **W2: Architectural Changes and Fairness of Baseline Comparison**
> >
> > **(1) Clarifying the architectural substitution and the use of SigLIP**
> >
> > Regarding the reviewer’s request for clarification on how SigLIP[1] is used, our implementation follows the official FLUX-Redux[2] prior, whose conditioning module is built upon SigLIP-ViT-L/14. SigLIP is a vision transformer trained on large-scale image–text pairs using a sigmoid contrastive loss, which yields more stable and semantically consistent image embeddings than the softmax-based CLIP objective. In the FLUX.1-Fill family, SigLIP-ViT-L/14 is the recommended image prior encoder for generating conditioning embeddings. Accordingly, in our system, the SigLIP features produced by the FLUX-Redux prior replace the text-token embeddings originally provided by T5, resulting in a purely image-conditioned inpainting model. We will further clarify this in the paper.
> >
> > **(2) Ensuring fairness of comparison**
> >
> > Concerning fairness of comparison, we performed an explicit control experiment using a standard DiT inpainting model with the same architecture, training pipeline, and SigLIP-based conditioning, with the only difference being the inclusion or removal of ESA. This evaluation isolates the contribution of ESA and avoids confounding factors related to backbone strength. The results show that ESA significantly outperforms the standard DiT baseline under identical settings, demonstrating that the gains do not arise from the modified backbone but from ESA itself. We will clarify this controlled comparison in the revised manuscript.
> >
> > ### Quantitative results on 2D-edits and 3D-edits
> > We report seven metrics for image quality, consistency, and editing effectiveness.
> > Best results are in **bold**, second best are *underlined*.
> >
> > ---
> >
> > ## **2D-edits**
> >
> > | Method                 | FID↓ | DINOv2↓ | KD↓ | SUBC↑ | BC↑ | WE↓ | MD↓ |
> > |------------------------|------|---------|-----|-------|------|------|------|
> > | RegionDrag             | 41.88 | 257.43 | 0.052 | 0.796 | $\underline{0.972}$ | 0.120 | 32.75 |
> > | MotionGuidance         | 146.41 | 1307.90 | 0.078 | 0.452 | 0.714 | 0.260 |145.46 |
> > | DragDiffusion          | 37.68 | 242.52 | 0.051 | 0.776 | 0.969 | 0.177 | 34.78 |
> > | Diffusion Handles      | 69.34 | 588.58 | 0.054 | 0.725 | 0.857 | 0.180 | 40.94 |
> > | GeoDiffuser            | 38.22 | 198.58 | 0.052 | 0.761 | 0.937 | 0.166 | 34.94 |
> > | DesignEdit             | 32.55 | 142.45 | 0.052 | 0.874 | 0.962 | 0.098 | 10.15 |
> > | Magic Fixup            | $\underline{27.32}$ | 114.08 | $\underline{0.051}$ | 0.889 | 0.966 | 0.075 | 10.39 |
> > | Standard DiT Inpainting| 28.88 | 113.38 | 0.052 | 0.902 | 0.971 | 0.064 | 10.51 |
> > | FreeFine               | 27.48 | $\underline{109.23}$ | 0.052 | $\underline{0.906}$ | 0.971 | $\underline{0.056}$ | $\underline{9.42}$ |
> > | **GeoEdit (Ours)**     | **25.07** | **90.66** | **0.051** | **0.910** | **0.977** | **0.054** | **9.23** |
> >
> > ## **3D-edits**
> >
> > | Method                 | FID↓ | DINOv2↓ | KD↓ | SUBC↑ | BC↑ | WE↓ | MD↓ |
> > |------------------------|------|---------|-----|-------|------|------|------|
> > | Diffusion Handles      |126.24 |1028.60  |0.056  |0.737  |0.885  |0.189  | $\underline{18.56}$ |
> > | GeoDiffuser            |77.34  |475.62   |0.055  |0.802  |0.946  |0.179  | 43.51 |
> > | Standard DiT Inpainting|68.02  |372.68   |0.055  |0.823  |0.962  |0.082  | 22.50 |
> > | FreeFine               |$\underline{65.94}$|$\underline{366.39}$|$\underline{0.055}$|$\underline{0.832}$|$\underline{0.967}$|$\underline{0.052}$| 21.27 |
> > | **GeoEdit (Ours)**     |**64.30**|**350.69**|**0.054**|**0.840**|**0.977**|**0.051**| **18.08** |
> >
> > ---
> >
> > **Reference**
> >
> > [1] Xiaohua Zhai, Basil Mustafa, Alexander Kolesnikov, and Lucas Beyer. Sigmoid loss for language image pre-training. In Proceedings of the IEEE/CVF international conference on computer vision, pp. 11975–11986, 2023.
> >
> > [2] Black Forest Labs. FLUX.1-Redux [dev] model card. https://huggingface.co/black-forest-labs/FLUX.1-Redux-dev, 2024.

---

> ### Author Response · Authors · 2025-11-27
>
> Dear Reviewer YQsq,
>
> Thank you again for your thoughtful review and valuable comments on our paper. We have updated our rebuttal and follow-up responses with additional clarifications and experiments to address your concerns, and we would be very grateful if you could take a moment to have a quick look and, if appropriate, consider updating your evaluation. Thank you very much for your time and effort in reviewing our work.

---

### Official Review · Reviewer_4ZEL · 2025-10-29

**Soundness:** 3
**Presentation:** 3
**Contribution:** 3
**Rating:** 8
**Confidence:** 3

**Summary:**

This paper proposes GeoEdit, a diffusion-transformer–based framework for geometric image editing.
Two key components are introduced: (1)Geometric Transformation module – performs translation, rotation, and scaling in a 3D-aware way via object reconstruction. (2)Effects-Sensitive Attention (ESA) – a soft guidance mechanism that modulates attention logits to preserve lighting and shadow realism, theoretically supported by a KL-divergence analysis.

The authors also construct RS-Objects, a dataset of 120k rendered + synthetic image pairs designed for geometric transformations and visual-effects learning.Extensive experiments on GeoBench show consistent quantitative and qualitative improvements over strong baselines such as FreeFine, GeoDiffuser, and Diffusion Handles.

**Strengths:**

1 Clear motivation and problem definition – focuses on geometric (translation / rotation / scaling) image editing, which remains under-explored compared to semantic or text-guided edits.

2 Comprehensive dataset pipeline – the RS-Objects dataset seems carefully designed (render + AIGC + human), addressing a genuine data gap.

3 Strong experiments – covers both 2D and 3D edits, reports seven metrics, and includes ablations and a user study

4 Readable writing and solid figures

**Weaknesses:**

1 Dataset authenticity & reproducibility. whether any real photographs with ground-truth geometric edits exist for validation. Public release status is unclear

2 Many baselines are test-time or training-free. The proposed method is training-based with a large custom dataset, so the comparison is not entirely apples-to-apples. how much overhead does ESA add versus standard DiT inpainting?

3 Limitations are not discussed, what is the thing that GeoEdit can not achieve? Discussing these aspects is crucial for guiding future research and fair benchmarking.

**Questions:**

The proposed ESA biases attention distribution statistically rather than modeling physical light transport. So, can GeoEdit guarantee physically correct shadow direction, reflection geometry, or color temperature consistency, especially when global illumination changes?

---

> ### Author Response · Authors · 2025-11-20
>
> We would like to sincerely thank the Reviewer for the insightful and detailed comments.
>
> **W1: Dataset Validity & Reproducibility Concerns**
>
> We thank the reviewer for the insightful comment. In addition to our main experiments, we also evaluate our method on the public ObjectMover benchmark [1], which includes real photographs with ground-truth geometric edits. Both qualitative and quantitative results, provided in Appendix I, consistently demonstrate the effectiveness of our approach. For completeness and reproducibility, we will release both the dataset and the code.
>
> **W2: Concerns on Baseline Comparability and Training Overhead**
>
> We thank the reviewer for the thoughtful comment.
> To ensure a fully apples-to-apples evaluation, we additionally test a standard DiT inpainting model whose training pipeline, architecture, and optimization settings are strictly identical to ours, with the only difference being the inclusion or removal of the ESA module. This design isolates the contribution of ESA itself. Experimental results show that ESA consistently outperforms the standard DiT baseline under the same architecture and training conditions, demonstrating that ESA provides a clear and independent improvement.
>
> ### Quantitative results on 2D-edits and 3D-edits
> We report seven metrics for image quality, consistency, and editing effectiveness.
> Best results are in **bold**, second best are $\underline{underlined}$.
>
> ---
>
> ## **2D-edits**
>
> | Method                 | FID↓ | DINOv2↓ | KD↓ | SUBC↑ | BC↑ | WE↓ | MD↓ |
> |------------------------|------|---------|-----|-------|------|------|------|
> | RegionDrag             | 41.88 | 257.43 | 0.052 | 0.796 | $\underline{0.972}$ | 0.120 | 32.75 |
> | MotionGuidance         | 146.41 | 1307.90 | 0.078 | 0.452 | 0.714 | 0.260 |145.46 |
> | DragDiffusion          | 37.68 | 242.52 | 0.051 | 0.776 | 0.969 | 0.177 | 34.78 |
> | Diffusion Handles      | 69.34 | 588.58 | 0.054 | 0.725 | 0.857 | 0.180 | 40.94 |
> | GeoDiffuser            | 38.22 | 198.58 | 0.052 | 0.761 | 0.937 | 0.166 | 34.94 |
> | DesignEdit             | 32.55 | 142.45 | 0.052 | 0.874 | 0.962 | 0.098 | 10.15 |
> | Magic Fixup            | $\underline{27.32}$ | 114.08 | $\underline{0.051}$ | 0.889 | 0.966 | 0.075 | 10.39 |
> | Standard DiT Inpainting| 28.88 | 113.38 | 0.052 | 0.902 | 0.971 | 0.064 | 10.51 |
> | FreeFine               | 27.48 | $\underline{109.23}$ | 0.052 | $\underline{0.906}$ | 0.971 | $\underline{0.056}$ | $\underline{9.42}$ |
> | **GeoEdit (Ours)**     | **25.07** | **90.66** | **0.051** | **0.910** | **0.977** | **0.054** | **9.23** |
>
> ---
>
> ## **3D-edits**
>
> | Method                 | FID↓ | DINOv2↓ | KD↓ | SUBC↑ | BC↑ | WE↓ | MD↓ |
> |------------------------|------|---------|-----|-------|------|------|------|
> | Diffusion Handles      |126.24 |1028.60  |0.056  |0.737  |0.885  |0.189  | $\underline{18.56}$ |
> | GeoDiffuser            |77.34  |475.62   |0.055  |0.802  |0.946  |0.179  | 43.51 |
> | Standard DiT Inpainting|68.02  |372.68   |0.055  |0.823  |0.962  |0.082  | 22.50 |
> | FreeFine               |$\underline{65.94}$|$\underline{366.39}$|$\underline{0.055}$|$\underline{0.832}$|$\underline{0.967}$|$\underline{0.052}$| 21.27 |
> | **GeoEdit (Ours)**     |**64.30**|**350.69**|**0.054**|**0.840**|**0.977**|**0.051**| **18.08** |
>
> ---
> Besides, ESA does not introduce any additional inference branches or large architectural modifications. When training or inferring at the same resolution, the memory usage of ESA increases by only 4–5 GB compared to standard DiT. This small increase mainly comes from the region-guidance mechanism used by ESA. The overhead remains within a practical and deployable range and does not affect applicability.
>
> **Reference**
>
>  [1] Xin Yu, Tianyu Wang, Soo Ye Kim, Paul Guerrero, Xi Chen, Qing Liu, Zhe Lin, and Xiaojuan Qi.
> Objectmover: Generative object movement with video prior. In CVPR, 2025.

---

> ### Author Response · Authors · 2025-11-20
>
> **W3: Limitations Analysis**
>
> We appreciate the reviewer’s attention to the method’s limitations. GeoEdit may struggle in scenes involving strong secondary physical effects. For example, when a motorcycle kicks up sand or dust, relocating the object may not fully transfer these extended effects to the new position. These dynamic, particle-based phenomena are fundamentally different from illumination-related effects (e.g., shading or soft shadows), which GeoEdit is specifically designed to handle. We will include this limitation in the final version.
>
> **Q: Discussion of Lighting Consistency**
>
> We thank the reviewer for raising this point. Our training data are constructed to provide correct physical lighting cues: the synthetic dataset is rendered in Blender with physically based rendering to ensure accurate shadow directions, reflection geometry, and color temperature, and the composited data are strictly human-filtered (details in Appendix C) to retain only physically consistent samples. This enables GeoEdit to learn physically plausible lighting behavior implicitly, even though ESA does not explicitly model light transport.
> As shown in Appendix I, we further evaluate GeoEdit on the ObjMove-A dataset [1], which contains diverse illumination conditions together with ground-truth edited results. As reported in Table 8, our method achieves the best performance across multiple metrics. Moreover, in the second-to-last row of Figure 11, when the tiger toy is moved from a shaded region into direct light, the model not only adjusts the object’s brightness according to the surrounding illumination but also produces a shadow that most closely matches the ground truth.
>
>
> **Reference**
>
> [1] Xin Yu, Tianyu Wang, Soo Ye Kim, Paul Guerrero, Xi Chen, Qing Liu, Zhe Lin, and Xiaojuan Qi.
> Objectmover: Generative object movement with video prior. In CVPR, 2025.

---

> ### Author Response · Authors · 2025-11-27
>
> Dear Reviewer 4ZEL,
>
> Thank you again for your thoughtful review and valuable comments on our paper. We have updated our rebuttal and follow-up responses with additional clarifications and experiments to address your concerns, and we would be very grateful if you could take a moment to have a quick look and, if appropriate, consider updating your evaluation. Thank you very much for your time and effort in reviewing our work.

---

### Author Response · Authors · 2025-12-01
**Overview of Reviews and Subsequent Discussion**

We thank the reviewers and the area chair for their time and constructive feedback. We would like to briefly summarize the reviews and the subsequent discussion, as this context may be helpful for your decision. In this work, we introduce **GeoEdit**, an in-context inpainting framework for geometric image editing under large object transformations in complex scenes, together with an **Effects-Sensitive Attention (ESA) module** that jointly handles geometry and lighting/shadow consistency, and we construct **RS-Objects**, a **large-scale** geometric editing dataset with **over 120K** high-quality rendered and synthesized image pairs that enables effective training and evaluation of such edits. These contributions form the technical basis for the reviewers’ assessments summarized below.

**Reviewer 4ZEL (score 8)** is broadly satisfied with the submission. They view our problem formulation (geometric image editing via in-context inpainting), the ESA module, and the RS-Objects dataset as clear and valuable contributions, and they **recommend poster acceptance.** Their comments mostly ask for clarifications and minor improvements (e.g., a clearer limitations section and a short discussion of computational cost), which we are happy to incorporate in the camera-ready version.

**Reviewer YQsq (score 4)** raises concerns that are largely centered around the lack of theoretical insight and a clearer justification of ESA. In the rebuttal, we **substantially expanded the theoretical analysis and added new experiments**, including ESA hyperparameter sensitivity, additional ablations, and comparisons under a controlled backbone setting. There was relatively little back-and-forth after this extended response, so we did not have the opportunity to discuss these additions in depth with this reviewer. Overall, we believe that many of their concerns were triggered by the **initial, compact theory section rather than by fundamental flaws in the method itself.**

**Reviewer XX6q (initially score 4, then indicated an update to 6)** appears to be **very knowledgeable in this research area** and engaged with the paper in considerable depth. They raised several technically sharp questions—for example, about ESA’s behavior with respect to lighting and shadows, its dependence on external 3D priors, backbone fairness, and failure cases—and even inspected our figures at high resolution to identify subtle spatial misalignments in some examples. After an extensive, **point-by-point discussion** in which we provided **new experiments and clarifications**, this reviewer stated that **their main concerns had been largely addressed** and that they would **raise their score to 6.** Importantly, their first major concern about ESA and its theoretical grounding is conceptually similar to a key issue raised by Reviewer YQsq, and our additional analysis and experiments directly target this shared point.

In summary, the **most critical and technically expert-leaning reviewer (XX6q)**, after a **detailed technical exchange**, is now moderately positive and explicitly **increasing their score**, while **Reviewer 4ZEL**, who initially gave an **8**, is already **supportive**. We hope this indicates that the main concerns can be adequately resolved with the clarifications and additional results we have provided, and that the paper can be considered favorably in the final decision.

---

### Meta-Review · Area_Chair_jerN · 2025-12-30

**Summary:**

The paper initially received two negative and one positive ratings. The concerns are mostly about 1) baseline comparisons with fairness, 2) modeling of physical constraints like lighting condition, 3) some technical clarifications, 4) failure cases and limitations, e.g., whether the proposed dataset is not generalizable, 5) more analysis, e.g., sensitivity to hyperparameters.

**Reviewer Concerns:**

The authors have provided responses in the rebuttal to answer initial concerns from the reviewers. The AC took a close look at the paper, reviews, and the rebuttal. After the rebuttal, the AC finds that most questions are addressed well, especially the further clarification of technical contributions and additional experiments for baseline comparisons and analysis. This also leads to the increased rating from reviewer XX6q to be positive. Therefore, the AC agrees with the reviewers' overall feedback and hence recommends the acceptance rating, while strongly encouraging the authors to revise the paper accordingly and release the dataset and code as promised.

**Reviewer Scores:**

Reviewer XX6q mentioned raising the original rating from 4 to 6, while the other two reviewers did not fully participate in the discussion.

---

### Decision · Program_Chairs · 2026-01-26

Accept (Poster)